# Why Do We Need Weight Decay in Modern Deep Learning?

**Francesco D'Angelo**[*], **Maksym Andriushchenko**[*], **Aditya Varre, Nicolas Flammarion**
Theory of Machine Learning Lab
EPFL, Lausanne, Switzerland
{francesco.dangelo,maksym.andriushchenko,aditya.varre,nicolas.flammarion}@epfl.ch

## Abstract

Weight decay is a broadly used technique for training state-of-the-art deep networks from image classification to large language models. Despite its widespread usage and being extensively studied in the classical literature, its role remains poorly understood for deep learning. In this work, we highlight that the role of weight decay in modern deep learning is different from its regularization effect studied in classical learning theory. For deep networks on vision tasks trained with multipass SGD, we show how weight decay modifies the optimization dynamics enhancing the ever-present implicit regularization of SGD via the *loss stabilization mechanism*. In contrast, for large language models trained with nearly one-epoch training, we describe how weight decay balances the *bias-variance tradeoff* in stochastic optimization leading to lower training loss and improved training stability. Overall, we present a unifying perspective from ResNets on vision tasks to LLMs: weight decay is never useful as an explicit regularizer but instead changes the training dynamics in a desirable way. The code is available at https://github.com/tml-epfl/why-weight-decay

## 1 Introduction

The training of modern neural networks broadly falls into two regimes: *over-training*, which involves multiple passes through a dataset and necessitates effective regularization strategies to avoid overfitting; and *under-training*, characterized by fewer passes due to large amounts of training data and computational constraints (Hoffmann et al., 2022). Modern deep learning unequivocally embodies both training regimes: ResNet architectures on computer vision tasks (He et al., 2016) serve as quintessential examples of the over-training regime, while the training of large language models (Brown et al., 2020) stands as a hallmark of the under-training regime. Despite their differences, both regimes extensively adopt weight decay as a regularization technique, though its effectiveness and role remain subjects of ongoing debate. For the first regime, Zhang et al. (2016) showed that even when using weight decay, neural networks can still fully memorize the data, thus questioning its regularization properties. For the second, regularization is inherently unnecessary as the limited number of passes already prevents overfitting. These considerations raise important questions about the necessity and purpose of weight decay, introducing uncertainty about its widespread usage. To illustrate the effect of weight decay in the two regimes,

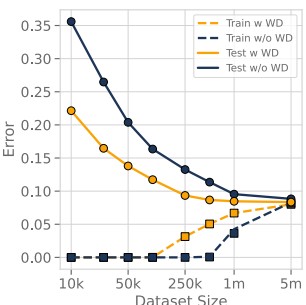

Figure 1: Test error vs. dataset size on CIFAR-10-5m for a *fixed* number of training iteration. Weight decay is helpful in both: the over-training and the under-training, one-pass regime.

---

[*]Equal contribution

we conduct a simple experiment. We train a ResNet18 on subsets of the CIFAR-5m dataset (Nakkiran et al., 2020) with sizes from $10\,000$ to 5 mln. The computational budget of each training session is fixed to 5 mln iterations, which amounts to a range of passes between $500$ and one. In the over-training regime (left in Fig. 1), weight decay does not prevent the models from achieving zero training error, but its presence still improves the test error. Attempting to explain this generalization benefit, recent works (Li & Arora, 2019; Li et al., 2020) bring forth the hypothesis that it is inadequate to think about weight decay as a capacity constraint since it still bears an effect on the training of scale-invariant models. As a result, understanding the effect of weight decay on the optimization dynamics becomes crucial to understanding generalization. Nevertheless, this line of work heavily relies on an *effective learning rate* (ELR) which only emerges as a consequence of scale-invariance and therefore does not apply to general architectures. In the under-training regime (right in Fig. 1), where the generalization gap vanishes, weight decay seem to facilitate faster training for slightly better accuracy. However, a characterization of the mechanisms through which weight decay impacts the training speed in this regime remains underexplored.

Our work delves into the mechanisms underlying the benefits of weight decay by training established machine learning models in both regimes: ResNet on popular vision tasks (over-training) and Transformer on text data (under-training). Towards this goal, we make the following contributions:

- In the over-training regime, we unveil the mechanism by which weight decay effectively reduces the generalization gap. We demonstrate that combining weight decay with large learning rates enables non-vanishing SGD noise, which through its implicit regularization controls the norm of the Jacobian leading to improved performance. Moreover, our investigation offers a thorough explanation for the effectiveness of employing exponential moving average and learning rate decay in combination with weight decay.

- In the under-training regime, particularly for LLMs trained with one-pass Adam, we confirm experimentally that weight decay does not bring any regularization effect and is simply equivalent to a modified ELR. We explain the training curves commonly observed with weight decay: through this ELR, weight decay better modulates the bias-variance trade-off, resulting in lower loss. Additionally, we show that weight decay has another important practical benefit: enabling stable training with the `bfloat16` precision.

## 1.1 Related work

The concept of employing $\ell_2$ weight penalty traces back to studies on the stability of solutions for ill-posed problems (Tikhonov, 1943). It has since been extensively explored in statistics (Foster, 1961; Hoerl, 1962; Hoerl & Kennard, 1970). Krogh & Hertz (1991) present one of the earliest systematic studies on weight decay tailored for *neural networks*. Generalization bounds, such as those by Shalev-Shwartz & Ben-David (2014), suggest that weight decay can be *sufficient* for generalization, although not necessary, e.g., due to the implicit regularization of gradient methods (Soudry et al., 2018). Zhang et al. (2016) argue that while weight decay improves test accuracy, the improvement is not substantial ($\approx$ 1-2% on ImageNet), indicating the key role of implicit regularization. Loshchilov & Hutter (2019) highlight the distinct effects of weight decay and $\ell_2$ regularization, particularly for Adam, suggesting that Adam combined with weight decay (AdamW) leads to superior regularization and simpler hyperparameter tuning. For GPT-3 training, Brown et al. (2020) suggest that they include weight decay to provide *a small amount of regularization*, although we believe it is not the primary reason as we discuss in Sec. 3.

Multiple works have focused on weight decay as a tool influencing optimization dynamics. Van Laarhoven (2017) emphasizes that weight decay's impact on scale-invariant networks is primarily seen in terms of an effective learning rate. Zhang et al. (2018) propose three mechanisms of weight decay regularization: (1) increasing the effective learning rate for scale-invariant networks, although as we discuss, the same holds for networks beyond scale invariance (2) approximating the regularization of the input Jacobian for an optimizer inspired by second-order methods, (3) inducing a specific dampening effect in this optimizer. Li & Arora (2019); Li et al. (2020) explore the optimization properties of scale-invariant deep networks for which the effective learning rate can be derived. Lewkowycz & Gur-Ari (2020) suggest that the best generalization is achieved with the smallest $\lambda$ although it necessitates longer training. Additionally, Lewkowycz (2021) propose a criterion for detecting when to decay the learning rate based on the evolution of the weight norm. Bjorck et al. (2021) explore the effect of decoupling weight decay, especially in the early stage of training. Li et al. (2022a) make BERT architecture scale-invariant to enhance training stability and make it more

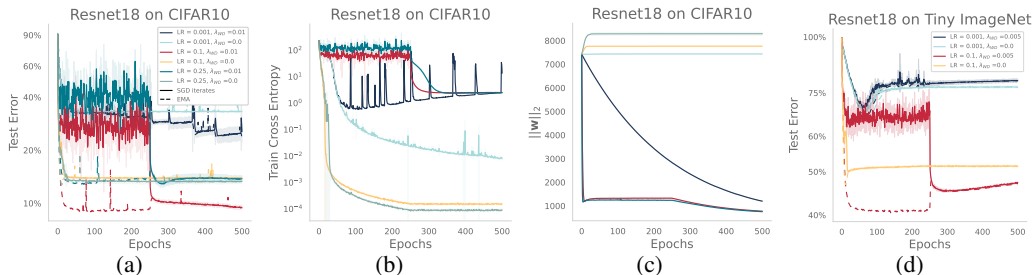

Figure 2: **Training with and w/o weight decay.** We report the test error for Resnet18 on CIFAR-10 (2a) and Tiny-ImageNet (2d) trained with and without weight decay and with small and large learning rates. We also include the correspondent EMA, represented by dashed lines. After the first 250 epochs the learning rate is decayed to $\eta = 10^{-3}$ for all the curves. We report also the L2 norm of the parameters (2c) and Train CE (2b) which after the decay converges to the same value for all the runs with the same $\lambda$.

compatible with standard SGD. Recently, Kosson et al. (2023) show a mechanism through which weight decay balances rotational updates across different layers that motivates a new optimizer.

The seminal paper of Krizhevsky et al. (2012) that introduced AlexNet suggest that weight decay serves not only as a regularizer but also reduces the model's training error, functioning as an *optimization tool*. In recent work, Hoffmann et al. (2022) briefly observe that weight decay enhances the training performance of Adam for training LLMs, but only after $\approx 80\%$ of the total iterations. However, they do not provide an explanation for this behavior, a point we delve into in Sec. 3.

## 2 Weight decay in the over-training regime

In this section, we delve into the influence of weight decay in the over-training regime, with a specific focus on image classification tasks. We focus on the training of ResNet models (He et al., 2016) using SGD on Tiny-ImageNet (Wu et al., 2017) and report additional experiments in appendix C for VGG, Resnet32 and scale-invariant Resnet architectures on CIFAR10 and CIFAR100 (Krizhevsky & Hinton, 2009).

**Notations and setup.** Let $(x_i, y_i)_{i=1}^n$ be the training inputs and labels where $x_i \in \mathbb{R}^d$, $y_i \in \mathbb{R}^c$, and $c$ is number of classes. Let $h : \mathbb{R}^p \times \mathbb{R}^d \to \mathbb{R}^c$ be the hypothesis class of neural network and for any parameter $\mathbf{w} \in \mathbb{R}^p$ where the function $h_{\mathbf{w}}(\cdot) : \mathbb{R}^d \to \mathbb{R}^c$ represents the network predictions. The training loss $\mathcal{L}$ and the $\ell_2$-regularized loss $\mathcal{L}_\lambda$, for $\lambda \geq 0$, are given by:

$$\mathcal{L}(\mathbf{w}) = \frac{1}{N} \sum_{i=1}^{N} \ell\left(y_i, h_{\mathbf{w}}(x_i)\right) \quad \mathcal{L}_\lambda(\mathbf{w}) = \mathcal{L}(\mathbf{w}) + \frac{\lambda}{2} \|\mathbf{w}\|^2.$$

where $\ell(\cdot, \cdot) : \mathbb{R}^c \times \mathbb{R}^c \to \mathbb{R}$ denotes the cross-entropy (CE) loss function. With $i_t \sim \mathbb{U}([N])$, the SGD algorithm on $\mathcal{L}_\lambda$ (here with batch size 1 and with replacement) with a learning rate (LR) $\eta$ is

$$\mathbf{w}_{t+1} = \mathbf{w}_t - \eta \nabla_{\mathbf{w}} \ell\left(y_{i_t}, h(\mathbf{w}_t, x_{i_t})\right) - \eta \lambda \mathbf{w}_t. \tag{1}$$

Along the training we track three different iterates: (1) **large-LR** denoted by $\mathbf{w}_t$ which use a large constant LR to exploit the SGD noise, (2) **fine-tuning** $\tilde{\mathbf{w}}_t$ which starting from $\mathbf{w}_t$ use a small LR and (3) the **exponential moving average** (EMA) $\bar{\mathbf{w}}_t$ along the large-LR iterates.

### 2.1 Loss stabilization and weight decay

To understand whether minimizing the regularized objective $\mathcal{L}_\lambda$ alone ensures optimal generalization, we compare test errors in Fig. 2a across various settings. Although both large and small LRs minimize the regularized objective, the evidence that optimal performance is achieved exclusively with large LRs indicates that *the objective alone is insufficient to explain the benefits of WD or ensure generalization.*[2] This experiment reaffirms the widely acknowledged consensus that *implicit regularization induced by the LR is crucial* (Keskar et al., 2016; Li et al., 2019; Andriushchenko et al.,

---

[2]The red, blue and green curves have the same CE 2b and norm 2c, hence same $\mathcal{L}_\lambda$ but different test error 2a.

2023). Despite revealing an interplay between weight decay and large initial LR, the understanding of the corresponding dynamics remains limited. In this section, our goal is to comprehensively understand these dynamics, particularly to elucidate the difference in generalization between training with and without weight decay and using different learning rates, as observed in Fig. 2a. Given the regularization of the $\ell_2$ norm of parameters, it is natural to wonder whether weight decay's improvement primarily stems from its ability to control the norm of the trained model. The experiment in Fig. 2c clearly illustrates that distinct training trajectories, while resulting in the same final $\ell_2$ norm for parameters, can yield different levels of generalization stating that *the $\ell_2$-norm of the learned model's parameters is inconsequential*. This observation suggests that once the norm is constrained by weight decay, the critical factor influencing the model's generalization is the subsequent choice of LR. We should note that neural networks can be explicitly made scale-invariant by means of normalization layers and small architectural changes. Li & Arora (2019) have used this setting to reveal that the training dynamics has an effective learning rate which, depending on the L2-norm of the parameters, reduces the effect of WD to merely a scheduler for the learning rate. Our analysis presents a broader perspective that does not depend on scale invariance. At the core of our examination are the unique properties of exponentially tailed loss functions, such as CE: when the data is separable and WD is not applied, the infimum of the loss is at infinity, leading to the unbounded growth of the weight norm (Ji & Telgarsky, 2019; Soudry et al., 2018). The application of WD, by inhibiting this growth, prevents the decrease of CE loss, which in turn, significantly alters the optimization dynamics.

Indeed, examining the parameter norm evolution in Fig. 2c, we notice how it rapidly decreases to stabilize within a small, approximately constant interval. Similarly, the Train CE in Fig. 2b displays a stabilization effect beyond which it cannot decrease without a reduction in the learning rate. We hypothesize that WD enables an optimization dynamic akin to that on the surface of a sphere of certain radius thus triggering the following essential mechanism:

*Constraining the parameter norm hinders the decrease of the CE, thereby enabling non-vanishing noise in SGD. This allows SGD implicit regularization to unfold and steer the optimization trajectory.*

Next, we empirically characterize this implicit regularization mechanism.

## 2.2   The noise driven process

The long-held belief that the implicit regularization property of SGD is pivotal to the generalization capabilities of Neural Networks has been a cornerstone in the field of deep learning (Keskar et al., 2016). Many theoretical studies (Blanc et al., 2020; Li et al., 2021b; Damian et al., 2021), attempting to understand this phenomenon, draw upon the essential finding that, in the case of regression and when Gaussian noise is added to the labels, the shape of the covariance of the stochastic gradients matches the shape of the Hessian. This allows Damian et al. (2021) and Pillaud-Vivien et al. (2022) to show that the trajectory of the SGD iterates closely tracks the solution of a regularized problem. In our analysis, we conjecture a similar result; the dynamics of SGD with CE, closely track a regularized process. The important difference in our statement is that unlike Blanc et al. (2020) and Damian et al. (2021), we do not need to add noise to the labels at each iteration. Instead, weight decay, in combination with large-LR induces a label noise-like behavior via loss stabilization (Andriushchenko et al., 2023). To better understand the interplay between weight decay, loss stabilization and the noise of SGD, it is convenient to consider the binary classification case where $y_i \in \{0, 1\}$. We also define the Jacobian of the network as $J(x_i, \mathbf{w}) := \nabla h_{\mathbf{w}}(x_i) \in \mathbb{R}^p$ and its norm averaged across the dataset: $\left\| J(\mathbf{w}) \right\|_F^2 = \frac{1}{N} \sum_{i=1}^{N} \text{Tr} \left( \nabla h_{\mathbf{w}}(x_i) \nabla h_{\mathbf{w}}(x_i)^\top \right)$. Denoting the noise of the gradient by $g_t = \nabla_{\mathbf{w}} \mathcal{L}(\mathbf{w}_t) - \nabla_{\mathbf{w}} \ell \left( y_{i_t}, h(\mathbf{w}_t, x_{i_t}) \right)$, the SGD update in equation 1 becomes:

$$\mathbf{w}_{t+1} = (1 - \eta\lambda)\mathbf{w}_t - \eta\nabla\mathcal{L}(\mathbf{w}_t) + \eta g_t \,. \tag{2}$$

Furthermore, we consider a Gaussian approximation of the SGD noise, matching the first and second moment of $g_t$. A substantial body of research has built upon this approximation and verified its validity (Li et al., 2020, 2021a; Smith et al., 2020; Xie et al., 2020; Li et al., 2021b). In particular Li et al. (2021a) demonstrated how modelling the SGD noise by a Gaussian is sufficient to understand its generalization. The mean is zero due to the unbiased mini-batch gradients: $\bar{g}(\mathbf{w}_t) := \mathbb{E}[g_t] = 0$

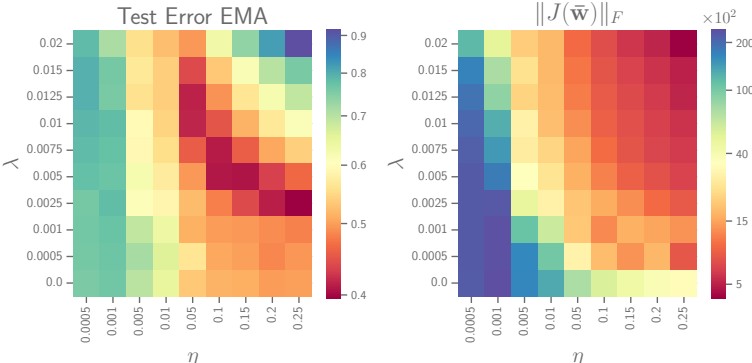

Figure 3: **Resnet18 on Tiny-ImageNet.** Heatmap of the test error and Jacobian norm for the EMA for all the different combinations of $\eta$ and $\lambda$.

whereas for the second moment:

$$
\Sigma_{\mathbf{w}_t} := \frac{1}{N}\sum_{i=1}^N \nabla\ell_i(\mathbf{w}_t)\nabla\ell_i^\top(\mathbf{w}_t) - \nabla\mathcal{L}(\mathbf{w}_t)\nabla\mathcal{L}^\top(\mathbf{w}_t) \approx \frac{1}{N}\sum_{i=1}^N \nabla\ell_i(\mathbf{w}_t)\nabla\ell_i^\top(\mathbf{w}_t)
$$

$$
\approx \frac{1}{N}\sum_{i=1}^N (\ell_i'(\mathbf{w}_t))^2 \nabla h_{\mathbf{w}_t}(x_i)\nabla h_{\mathbf{w}_t}(x_i)^\top \approx \frac{1}{N}\sigma_{\eta,\lambda}^2(\mathbf{w}_t)\sum_{i=1}^N \nabla h_{\mathbf{w}_t}(x_i)\nabla h_{\mathbf{w}_t}(x_i)^\top. \quad (3)
$$

In equation 3, we consider $\nabla\mathcal{L}(\mathbf{w}_t)\nabla\mathcal{L}^\top(\mathbf{w}_t)$ negligible compared to $\nabla\ell_i(\mathbf{w}_t)\nabla\ell_i^\top(\mathbf{w}_t)$ as the gradient noise variance dominates the square of the gradient noise mean. This fact has been used in previous works (Mori et al., 2022; Zhu et al., 2018; Jastrzebski et al., 2017) and in particular Jastrzebski et al. (2017) and Saxe et al. (2019) empirically verify it. Finally, we assume that the first derivative is approximately constant across all datapoints: $\ell_i'(\mathbf{w}_t) \approx \ell_j'(\mathbf{w}_t)\ \forall i,j$ and denote this common quantity as $\sigma_{\eta,\lambda}(\mathbf{w}_t)$. This last approximation is referred to as "decoupling approximation" and has been empirically verified for classification (Mori et al., 2022). Furthermore, in App. C.6 we performed additional experiments to verify the decoupling approximation by comparing the spectrum of the SGD covariance with and without this approximation during the large LR phase.

Altogether, these considerations lead to the following formulation of the SGD update:

$$
\mathbf{w}_{t+1} \approx (1 - \eta\lambda)\mathbf{w}_t - \eta\nabla\mathcal{L}(\mathbf{w}_t) - \frac{\eta}{N}\sigma_{\eta,\lambda}(\mathbf{w}_t)\sum_{i=1}^N \nabla h_{\mathbf{w}_t}(x_i)\,\xi_i^t, \qquad \text{where } \xi_i^t \sim \mathcal{N}(0,\mathbb{I}). \quad (4)
$$

This series of approximations, allows us to define the quantity $\sigma_{\eta,\lambda}(\mathbf{w}_t)$, which has a fundamental role in the characterization of the training dynamics. We refer to it as *the scale of the noise* because it regulates its intensity. Although $\eta$ and $\lambda$ influence the noise scale indirectly through the trajectory of $\mathbf{w}_t$, we explicitly highlight this dependence to emphasize our objective: to characterize the influence of WD and LR on the stochastic dynamics of SGD. We develop this characterization building upon the connection between the Jacobian of the network and the covariance of the SGD noise, which motivates the introduction of the following implicit regularization mechanism:

**Conjecture 1.** *Consider the algorithm in Eq. 1 with $\mathbf{w}_0$ initialized from a distribution $\mu_0\left(\mathbb{R}^{(p)}\right)$. For any input $x$, let $\mathbf{w}_t, h(\mathbf{w}_t, x)$ be the random variables that denote the iterate at time $t$ and its functional value. The stochastic process $(h(\mathbf{w}_t, x))_{t \in \mathbb{N}}$ converges to the stationary distribution $\mu_{\eta,\lambda}^\infty(x)$ with mean $\bar{\mu}_{\eta,\lambda}(x) = h\left(\mathbf{w}_{\eta,\lambda}^*, x\right)$ for which $\mathbf{w}_{\eta,\lambda}^*$ is a first-order stationary point of the following regularized loss:*

$$
\bar{\mathcal{L}}_\lambda(\mathbf{w}) := \mathcal{L}_\lambda(\mathbf{w}) + \eta\sigma_{\eta,\lambda}^2\left\| J(\mathbf{w})\right\|_F^2. \quad (5)
$$

The conjecture illustrates how varying noise levels correspond to distinct processes, wherein the mean of each solves a unique regularized problem. Moreover, it describes how each noise level determines the strength of the regularization. Using the mean to formulate the conjecture is a natural choice; even at stationarity,[3] the values of the loss $\mathcal{L}(\mathbf{w}_t)$ and the regularizer $\left\| J(\mathbf{w})\right\|_F^2$ would be dominated

---

[3]Assuming the existence of a stationary distribution, the iterates $\mathbf{w}_t$ are eventually realizations from it.

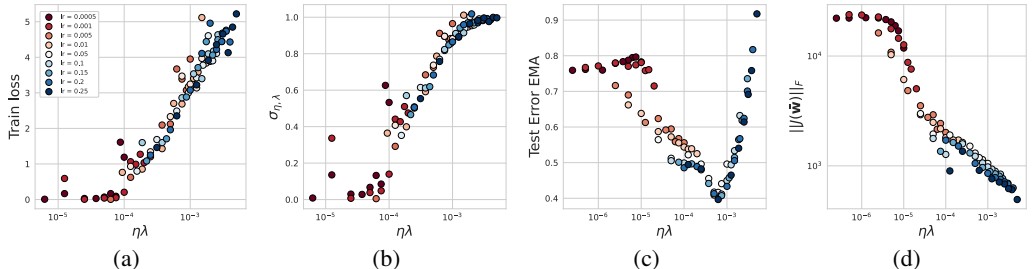

Figure 4: **Resnet18 on Tiny-ImageNet.** Training for 200 epochs with different $\eta$ and $\lambda$; the scale of the noise monotonically increases with the train loss and $\eta \times \lambda$ Fig. 4a, 4b. The test error instead, presents an optimal value of $\eta \times \lambda$ Fig. 4c while the Jacobian norm decreases monotonically Fig. 4d.

by the noise. To unveil the existence of an implicit regularization and to analyze the evolution of the distribution, we need to look at its summary statistics, in this case, the mean. While insights from Langevin dynamics suggest employing learning rate annealing to converge towards the mean of the stationary distribution, this approach introduces additional complexities which we discuss in Section 2.3. We instead consider an exponential moving average (EMA) $(\bar{\mathbf{w}}_t)_{t \geq 0}$ of the SGD iterates with parameter $\beta = 0.999$.

The most important implication of the conjecture is that the strength of the regularization $\sigma_{\eta,\lambda}$ depends on both the LR $\eta$ and the WD parameter $\lambda$. Our experiments in Fig. 3, provide empirical validation for this conjecture. When trained with different combinations of $\eta$ and $\lambda$, the EMA converges to models with different test performances. When fixing $\lambda$ there exists an optimal value of learning rate $\eta$ which gives the best test performance while the Jacobian norm monotonically increases. A similar picture can be drawn when fixing the learning rate.

Therefore, given two solutions $\bar{\mu}_{\eta_l,\lambda_l}$ and $\bar{\mu}_{\eta_s,\lambda_s}$ for which Test error$(\bar{\mu}_{\eta_l,\lambda_l})$ < Test error$(\bar{\mu}_{\eta_s,\lambda_s})$; the difference in their performances can be explained with the difference in their regularization strengths $\sigma_{\eta,\lambda}$. The solution $\bar{\mu}_{\eta_l,\lambda_l}$ benefits from better regularization and therefore endows better generalization properties. Furthermore, the heatmap for test error in Fig. 3 indicates that the minimum error is not achieved by a single combination of $\eta$ and $\lambda$, but rather along a contour where their product $\eta \times \lambda$ appears to be constant. This observation suggests an optimal trade-off between the learning rate and weight decay parameter, characterized by a curve in the parameter space where their product remains constant. Characterizing this relationship might reveal a useful tool which practitioners can adopt to optimally select values of weight decay and learning rate. Fig. 4c, 4d confirm that the product $\eta\lambda$ is the quantity controlling the regularization; combinations of $\eta$ and $\lambda$ with the same product show similar test performances and Jacobian norm. For Tiny-Imagenet, the test error exhibits an optimal value for $\eta\lambda \sim 0.005$ where increases beyond this point lead to *over-regularization* and decreases result in *under-regularization*. Simultaneously, the Jacobian norm $\|J\|_F$ consistently exhibits a monotonically decreasing trend.

To better understand the properties of the noise scale during training, we can observe in Fig. 4a, 4b how higher values of the training loss given by larger $\eta\lambda$ correspond to higher values of the noise scale $\sigma_{\eta,\lambda}$. The latter is measured by computing the Frobenius norm of the first derivative of the loss with respect to the predictions[4] averaged over all training datapoints $\frac{1}{N}\sum_{i=1}^{N}\|\ell'_i(\mathbf{w})\|_F$. The scale of the noise and therefore the strength of the regularization vanish when $\mathcal{L}(\mathbf{w}) \approx 0$ which happens for small values of $\eta\lambda$. Therefore, WD combined with a large LR stabilizes the CE to a larger value, preventing the noise from vanishing and regulating the implicit regularization. This fact further confirms Conjecture 1, demonstrating that with smaller noise scales, the mean (EMA) tends towards a point where the Jacobian's norm is higher, compared to trajectories with larger noise scales. To further validate our conjecture, we created snapshot ensembles in the ResNet18 on CIFAR-10 setting, by averaging in function space along the SGD trajectory every 10 epochs for the combinations of learning rate (LR) and weight decay (WD) considered. To assess whether the mean of the stationary distribution in function space aligns closely with the EMA, where the Jacobian norm is regularized, we compared the performance of snapshot ensembles with that of the EMA. Additionally, we computed the Total Variation Distance between the softmax outputs of the ensemble and the EMA. The results are reported in App. C.7 and show a strong alignment.

---

[4]In the multi-class setting $\ell'_i(\mathbf{w}_t) \in \mathbb{R}^c$; to quantify the scale, we compute its Frobenius norm.

**The role of the dynamics of the norm.** As discussed at the end of previous sub-section, after a rapid initial decrease of norm, the optimization resembles the dynamics of SGD projected onto a sphere. We stress that this is the crucial phase in training and the implicit regularization induced by SGD during this spherical optimization leads to better generalization. To validate this observation and isolate it from the initial norm drop, we investigate the behavior of SGD on a sphere with scale-invariant networks (Li & Arora, 2019). Scale invariance is chosen for its ease of LR tuning and for comparison with existing works (Kodryan et al., 2022; Li et al., 2020). Fig. 15 depicts a similar phenomenon as Fig. 3, where the test error vs. LR exhibits a U-shaped curve. While Kodryan et al. (2022) makes a similar observation, they do not provide an explanation. We demonstrate that the implicit regularization of the Jacobian norm is the key factor, elucidating its dependence on LR.

**Effective learning rate vs. high training loss.** Zhang et al. (2018); Van Laarhoven (2017) explored the relationship between LR and WD, introducing the concept of effective LR $\eta_e = \eta / \|\mathbf{w}\|_2^2$. These studies highlight that WD, preventing unbounded growth of the norm, enables the training process to evolve with a higher effective LR. This hypothesis is justified only with scale-invariance (which does not hold for general architecture). Furthermore, the underlying mechanism by which a higher LR enhances generalization is understood only in limited settings (Li et al., 2019). We propose that a high LR, combined with WD, leads to an increase in $\sigma_{\eta,\lambda}$. This hypothesis allows us to fully characterize and understand the mechanism through which generalization is enhanced.

**Mixing in the function space.** A simpler conjecture could have been stated in terms of the mixing of the iterates $(\mathbf{w}_t)_{t \geq 0}$ towards a solution of the regularized objective $\mathbf{w}_\eta^*$. However, Li et al. (2020) shows empirical evidence against mixing in the parameter space, emphasizing the necessity of considering the function space. Hence, our conjecture is formulated to capture stationarity in function space.

**On the benefit of normalization.** Our conjecture characterizes the mixing distribution but does not delve into the speed of the mixing process. In our experiments, we observe that normalization layers enable faster mixing. Li et al. (2020) observes a similar phenomenon in the case of scale-invariant networks, specifically the fast equilibrium conjecture, which is addressed by Li et al. (2022b). We note that this phenomenon persists even when the models are not exactly scale-invariant.

## 2.3 EMA and Fine-tuning

The large-LR phase sets the stage for SGD's inherent biases to emerge but to actually exploit such bias, reducing the stochastic noise is necessary. This can be attained in two different ways: averaging (EMA) or decaying the learning rate (fine-tuning), both strategies are widely adopted in practice. This section illustrates the relation between the two and highlights the benefits of using one over the other and the implications for our analysis. From a practical standpoint, implementing EMA is more efficient than LR-decay because it does not require additional gradient iterations or hyperparameter tuning. While both methods enhance performance, their effectiveness is contingent on being combined with loss stabilization, supporting the hypothesis that the noisy dynamics is the underlying factor for their success. Although EMA shows only a slight advantage, our experiments in Fig. 5b, 2a, 2d demonstrate that it consistently outperforms learning rate decay in various settings.

When empirically validating Conjecture 1, the EMA is useful to characterize the limit point (i.e., $t \to \infty$) but cannot adequately capture the dynamics throughout the entire trajectory. This limitation arises because different points along the trajectory are at different loss values, making the comparison of any relevant regularized quantities inconsistent. An approach to overcome this inconsistency is to project the iterate $\mathbf{w}_t$ onto a manifold of constant loss. This can be achieved via early-stopped gradient flow (Li et al., 2021b) (SGD with small LR) on the CE loss with $\lambda = 0$ where $\mathbf{w}_t$ is projected to a nearby point $\tilde{\mathbf{w}}_t$, such that $\mathcal{L}(\tilde{\mathbf{w}}_t) \sim \text{const.}, \forall t$. In practice, this corresponds to fine-tuning via LR-decay. Since after fine-tuning $\mathcal{L}(\tilde{\mathbf{w}}_t) \approx \mathcal{L}(\tilde{\mathbf{w}}_{t'}), \forall t, t'$ see Fig. 5a, we can compare $\|J(\tilde{\mathbf{w}}_t)\|_F$ and $\|J(\tilde{\mathbf{w}}_{t'})\|_F$ and understand its evolution. In the experiments detailed in Fig. 5c, we report $\|J\|_F$ along the fine-tuned iterates $\tilde{\mathbf{w}}_t$ and observe a decreasing trend, i.e., the sequence $\{\|J(\tilde{\mathbf{w}}_t)\|\}_{t \geq 0}$ is decreasing. This fact empirically validates that the entire trajectory of the iterates $(\mathbf{w}_t)_{t \geq 0}$, closely following the trajectory of the fine-tuned iterates $(\tilde{\mathbf{w}}_t)_{t \geq 0}$, bias the model towards a regularized solution that might enhances generalization. This also explains why learning rate schedules, such as step-decay, which starts with a large value and then decrease it, can enhance generalization.

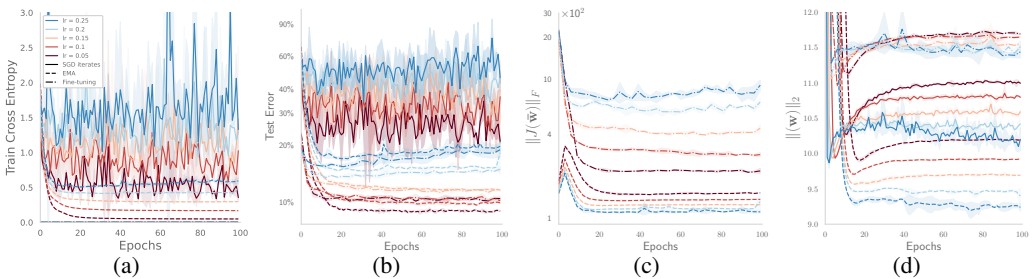

| | | | |
|---|---|---|---|
| (a) | (b) | (c) | (d) |

Figure 5: **EMA vs Fine-tuning.** Training of standard Resnet18 on CIFAR-10 for 100 epochs fixing $\lambda = 0.0125$ and varying the learning rate. In Fig. 5a we report different levels of loss stabilization, in Fig. 5b we report the test errors and in Fig. 5c and Fig. 5d the norm of the Jacobian and of the weights respectively. The quantities are measured for the SGD iterates, the EMA and the fine-tuning. The latter is performed for 100 epochs every 3 with $\eta = 10^{-3}$.

Despite providing a straightforward methodology to analyze the trajectory, LR-decay introduces additional complexities that cause deviations from the conjecture. Indeed, in Fig. 5c we observe that the final points of the fine-tuned iterates report the opposite trend compared to the EMA (smaller $\eta$ lead to larger $\|J\|_F$). This discrepancy is potentially due to the state-dependent nature of the SGD noise covariance in equation 3; decreasing the LR and removing WD can alter the stationary distribution and the regularized objective, leading to a different solution than anticipated by Conjecture 1.

## 3 Weight decay in the under-training regime

In this section, we investigate how WD enhances optimization in the under-training regime. Although the phenomenon is more general, we focus on LLMs for which one-epoch training is typically used.

**Two key effects of weight decay in the under-training regime.** WD is widely used in training state-of-the-art LLMs like GPT-3, Chinchilla, and Llama (Brown et al., 2020; Hoffmann et al., 2022; Touvron et al., 2023). While Brown et al. (2020) suggest that WD offers "*a small amount of regularization,*" its necessity remains unclear in the context of *one-pass* training where the population loss is directly minimized. As a sanity check, in Fig. 19 in Appendix, we verify that the generalization gap is close to zero even for models trained without WD. Instead of the regularization effect, we suggest that the two most crucial effects of WD in the under-training regime are (1) better optimization of the training loss as briefly observed by Hoffmann et al. (2022), (2) prevention of loss divergences under the `bfloat16` weight precision. We reproduce this phenomenon at a smaller scale with 124M parameters in Fig. 6: the final training loss is smaller for $\lambda$ equal to 0.1 and 0.3 compared to 0. We study both mechanisms which stand in contrast to the over-training regime of Sec. 2, where the primary concerns are not optimization and stability, but rather generalization.

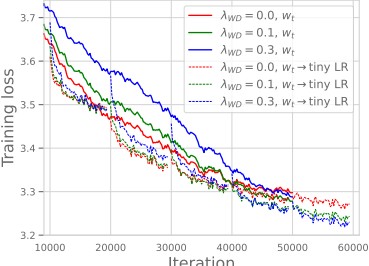

Figure 6: **GPT-2-124M on OpenWeb-Text.** We reproduce the improvement from WD as in Hoffmann et al. (2022) but at a much smaller scale. Performing fine-tuning with a tiny LR reveals that a higher starting training loss can still be a better point in terms of the final loss after fine-tuning.

**Experimental setup.** We use the `NanoGPT` repository (Karpathy, 2023) for training GPT-2 models (Radford et al., 2019) on OpenWebText. We train a 124M parameter model (known as GPT-2-Small) for 50 000 iterations with a batch size of 256. For most experiments, we reduce the default context length from 1024 to 256 to ensure practicality within an academic budget. Alternatively, we could have reduced the number of training iterations or batch size, but this would lead to insufficiently trained models. Unless mentioned otherwise, we train with AdamW using the default LR 0.0006, a short 400-iteration LR warmup, gradient clipping with the $\ell_2$-threshold 1.0, and $10\times$ cosine LR decay. We keep all other hyperparameters at their default values (see App. B).

**Better optimization with WD is reproducible at a smaller scale.** The findings from Hoffmann et al. (2022) (Fig. A7 therein) indicate that WD in AdamW leads to lower training loss ($\approx 0.02$ lower), primarily towards the end of training. The reduction of training loss directly translates to a better downstream performance and makes this observation practically relevant. Additionally, performing fine-tuning with a tiny LR reveals that a higher starting training loss can still be a better starting point

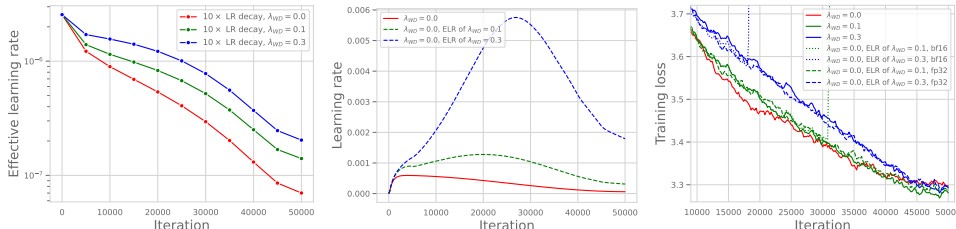

Figure 7: **GPT-2-124M on OpenWebText.** *Left*: The effective LR $\eta_t/\|\mathbf{w}_t\|_2$ for the models reported in Fig. 6. *Middle*: The LR schedule that matches the effective LR $\eta_t/\|\mathbf{w}_t\|_2$ of the runs with weight decay 0.1 and 0.3. *Right*: Matching the effective LR is sufficient to match the whole training dynamics of the loss if we avoid the loss spikes by using full precision (`float32` instead of `bfloat16`).

in terms of the final loss after fine-tuning. Moreover, in Fig. 21, we show that decoupling WD, as advocated by Loshchilov & Hutter (2019), is not necessary to achieve this effect: a simple $\ell_2$ penalty added to the loss suffices. Lastly, in Fig. 22, we show that a similar improvement in training loss is observed for *SGD with momentum* suggesting that adaptive LRs are not key for this phenomenon.

**Effective LR induced by weight decay in AdamW.** We hypothesize that the use of WD for LLM training results in an increased effective LR, even in the absence of scale invariance of the training loss for modern transformer architectures. Here we show that WD in combination with sign SGD— utilized as a surrogate for Adam (Balles & Hennig, 2018)—is equivalent to projected SGD on the sphere, with an effective LR $\eta_{\text{eff}} \propto \eta_t/\|\mathbf{w}_t\|_2$, similarly to Van Laarhoven (2017). Consider the update rule of sign SGD on loss $\ell$ with WD:

$$\mathbf{w}_{t+1} = (1 - \eta_t\lambda_t)\,\mathbf{w}_t - \eta_t \operatorname{sign}(\nabla\ell_t(\mathbf{w}_t)) = (1 - \eta_t\lambda_t)\,\|\mathbf{w}_t\|_2 \left[ \frac{\mathbf{w}_t}{\|\mathbf{w}_t\|_2} - \frac{\eta_t \cdot \operatorname{sign}(\nabla\ell_t(\mathbf{w}_t))}{(1 - \eta_t\lambda_t)\,\|\mathbf{w}_t\|_2} \right].$$

Considering the evolution of the direction $\tilde{\mathbf{w}} := \mathbf{w}/\|\mathbf{w}\|_2$,

$$\tilde{\mathbf{w}}_{t+1} \propto \left[ \tilde{\mathbf{w}}_t - \frac{\eta_t}{(1 - \eta_t\lambda_t)\,\|\mathbf{w}_t\|_2} \cdot \operatorname{sign}(\nabla\ell_t(\mathbf{w}_t)) \right].$$

When $\operatorname{sign}(\nabla\ell_t(\mathbf{w}_t))$ is determined solely by the direction $\tilde{\mathbf{w}}_t$, the evolution of the direction of weights becomes the primary matter. This scenario occurs when the function $\ell$ is scale-invariant or homogeneous. Observing the trend of the gradient norm and the parameter norm $\|\mathbf{w}_t\|_2$ from Fig. 26, we note an inverse relationship, i.e., the gradient norm is higher when the parameter norm is lower. This behavior is reminiscent of scale-invariant networks for which $\nabla\ell(\alpha\mathbf{w}) = \frac{1}{\alpha}\nabla\ell(\alpha\mathbf{w})$, for any $\alpha \neq 0$. Thus, controlling parameter norms with WD allows implicit changes to the LR schedule which we verify experimentally next.

**Matching the effective LR *without* weight decay.** To verify the key role of the ELR $\eta_t/\|\mathbf{w}_t\|_2$, we train a model without WD but with ELR corresponding to models trained with $\lambda \in \{0.1, 0.3\}$ and report results in Fig. 7. We observe that the *whole* training dynamics is matched which confirms our hypothesis. This fully explains the observation from Hoffmann et al. (2022) about the advantage of AdamW over Adam: there exists an LR schedule (albeit a non-standard one) shown in Fig. 7 (*middle*) that leads to the same loss profile as the original AdamW run. However, we note that this holds only for full `float32` precision, and models trained with `bfloat16` precision diverge in the middle of training. This experience suggests that WD is still necessary in practice to prevent loss divergence. We also note that matching the ELR $\eta_t/\|\mathbf{w}_t\|_2$ derived above for sign SGD instead of $\eta_t/\|\mathbf{w}_t\|_2^2$ for plain SGD (Zhang et al., 2018; Hoffer et al., 2018) is critical for AdamW. Otherwise, the runs diverge very early in training, even with `float32` parameter precision.

**Explaining the training dynamics of AdamW.** Classical optimization theory suggests that convergence of SGD-based algorithms primarily depends on two factors: the *bias* term that influences the rate at which initial conditions are forgotten and the *variance* term that results from noise in the gradient estimates (Moulines & Bach, 2011). We argue that these two factors, together with the observation about higher ELR for WD, can explain the loss profiles from Fig. 6. If we consider the simple case of SGD with a constant LR $\eta$ applied to a linear least-squares problem, the expected excess risk after $t$ iterations can be bounded as a sum of a bias and variance terms:

$$\text{Excess Risk} \lesssim (1 - \eta\mu)^t \|\mathbf{w}_0 - \mathbf{w}_*\|^2 + \eta\sigma^2,$$

where $\sigma$ is a uniform bound on the variance of the noise of gradient estimates, $\mu$ a lower bound on the objective function's Hessian, $\mathbf{w}_0$ the initial point and $\mathbf{w}_*$ the optimum. For linear models, it is well-established that a larger LR accelerates the contraction of the bias term but has a detrimental impact on the variance term, ultimately leading the variance term to dominate. Coming back to the dynamics in Fig. 6, with a large ELR at the start, the convergence becomes primarily bottlenecked by the high variance term proprtional to the learning rate, leading to higher loss values in the presence of WD. Conversely, towards the end of training, when ELR and the variance term are reduced, we see that WD catches up and performs better at the end, thanks to its relatively higher ELR *throughout the training* and thus better bias contraction. This perspective sheds light on the observation that EMA for LLMs is most advantageous when employed with large LRs (Sanyal et al., 2023) as we also illustrate in Fig. 23. As the variance dominates in this case, variance reduction of the averaging helps.

**Experiments with `bfloat16`.** Training in reduced precision is essential for speeding up training and reducing GPU memory requirements (Kalamkar et al., 2019). We further elaborate on the fact that WD is not fully equivalent to higher ELR and remains necessary for stable `bfloat16` training. While Scao et al. (2022) observe that usage of `float16` can cause loss divergences, `bfloat16` is considered much more stable and is de-facto standard in LLM training. Although `bfloat16` shares the same floating-point exponent size as `float32` (thus, the *range* of possible values is the same), it offers lower precision, with only 7 bits for the fraction instead of 23.

We observe that even more stable `bfloat16` data type can still exhibit late-training spikes that irreparably harm model performance *in standard practical settings*, such as with a larger context length (e.g., 1024 instead of 256 as in the previous experiments). Therefore, we focus on this configuration for the experiments shown in Fig. 8. Runs with a moderate LR 0.001 (the default LR of Adam) without WD exhibit late-training divergence for *all* random seeds when using `bfloat16`. By comparison, training with `float32` remains entirely stable. Importantly, we observe that the model *does not recover* after the loss spikes which contrasts with the loss spikes described in the Edge of Stability phenomenon (Cohen et al., 2021, 2022). We emphasize that all these runs use gradient clipping with the standard $\ell_2$-threshold. Finally, we observe that divergences can be prevented by reducing the LR, e.g., from 0.001 to 0.0006. However, this adjustment leads to slower training, as illustrated in Fig. 24 in the Appendix. Instead, the most effective approach is to use a higher LR of 0.001 *with WD*, which enables stable `bfloat16` training and yields a better final training loss.

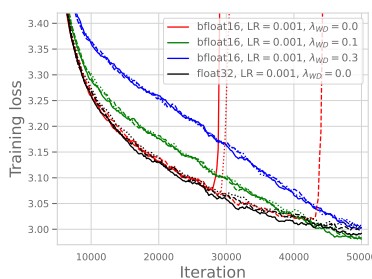

Figure 8: **GPT-2-124M on OpenWeb-Text with context length 1024.** Weight decay prevents divergence for LR 0.001 and enables stable `bfloat16` training. The three random seeds are denoted with —, - - -, · · · lines.

## 4    Conclusions

In this paper, we demonstrate how weight decay, a single hyperparameter, can manifest three distinct effects across different training regimes: it offers regularization when combined with stochastic noise, improves optimization of the training loss, and guarantees stability in low-precision training environments. In the over-training regime, the scale of the noise $\sigma_{\eta,\lambda}$ is the fundamental quantity governing the implicit regularization strength of SGD. Weight decay combined with large LR enables the noisy dynamics to evolve by maintaining the scale at a constant level. Techniques such as EMA or fine-tuning work by reducing noise, thereby allowing for the effective exploitation of the accumulated hidden regularization. Coming to the under-training regime, AdamW (Loshchilov & Hutter, 2019) was introduced as a *regularization* method. Instead, we argue that it is effective as it modulates the ELR to attune the bias-variance tradeoff. In addition, it also improves the stability of training. In summary, weight decay is seldom valuable as an explicit regularizer; instead, its widespread adoption can be attributed to its ability to induce desirable changes in optimization dynamics. We also acknowledge limitations of our work: given our limited computational resources, we do not conduct truly large-scale experiments. Moreover, we do not prove new theoretical results. Instead, we strive to provide a clear experimental picture and formulate general explanations for the effectiveness of weight decay in different training regimes.

## Acknowledgements

We thank Atli Kosson and Alex Damian for fruitful discussions and suggestions. M.A. was supported by the Google Fellowship and Open Phil AI Fellowship. A.V. was supported by the Swiss Data Science Center Fellowship. F.D. was supported by the Swiss National Science Foundation (grant number 212111)

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

# A  An additional comparison with related works

Our focus in Section 2 is on an empirical illustration of the implicit regularization phenomenon, hence we refrain from attempting to prove this general conjecture, which we believe is a challenging task. The existing theoretical works Blanc et al. (2020); Li et al. (2021b); Damian et al. (2021) present two major weaknesses; they are essentially limiting analysis and as such fail at capturing the *entire optimization trajectory* and they primarily target regression tasks. The powerful mathematical framework for scale-invariant networks developed by Li & Arora (2019); Li et al. (2020) allows them to study in detail the benefits of normalization and its interplay with weight decay. By means of this framework, they state a fast equilibrium conjecture, which gives qualitative guarantees for the speed of convergence of the stochastic process to the stationary distribution in function space. They disentangle the evolution of the norm and the direction of the parameters and show how the evolution of the direction only depends on the intrinsic LR $\lambda_i = \eta\lambda$. However, a qualitative description of the stationary distribution, its dependence on this intrinsic LR and the relationship with generalization is missing Li et al. (2020). We attempt to fill this gap by providing a qualitative depiction of the stationary distribution and its dependence on the intrinsic LR shading some light towards understanding the relationship with generalization. The work of Kodryan et al. (2022) reports a similar observation, where the best test loss is achieved at a LR where the loss neither converges nor diverges but does not provide any explanation.

Table 1: Comparison of our work with closely related works on regression and implicit regularization phenomenon induced by noise in the algorithm.

| Paper | Loss function | Algorithm | Implicit regularization |
|---|---|---|---|
| Damian et al. (2021) & Li et al. (2021b) | Squared loss & CE + label smoothing | Label noise GD | Trace of Hessian |
| Blanc et al. (2020) | Squared loss | Label noise GD | Jacobian norm |
| Li et al. (2020) | Scale-invariant loss | SGD | - |
| Andriushchenko et al. (2023) | Squared loss | SGD with large LR | Jacobian norm |
| Our work | Regularized CE | SGD with large LR | Jacobian norm |

# B  Training details

Full experimental details are available in our public repository `https://github.com/tml-epfl/why-weight-decay` but we also list the main training details here. All the experiments are conducted for 3 different random seeds, the error-bars report one standard deviation. **CIFAR-10/100 experiments.** We train a VGG network without BatchNorm and preactivation ResNet-18 on CIFAR-10 and ResNet-34 on CIFAR-100 without data augmentations. We use standard SGD *without momentum* for all experiments. We note that $\ell_2$ regularization and weight decay are exactly the same in this case. We use the standard He initialization (He et al., 2015) for all parameters. To make ResNets scale-invariant, we follow the approach of Li et al. (2020) consisting of fixing the last layer, removing the learnable parameters of the normalization layers and adding a normalization layer in the skip connection. For the experiments in Fig.11, VGG is trained with LR = 0.1 and LR = 0.01 and weight decay parameter is fixed to be either $\lambda = 0.0$ or $\lambda = 0.008$. The ResNet-18 is trained with LR = 0.08 and LR = 0.001 and $\lambda = 0.0$ or $\lambda = 0.0125$. The ResNet-34 is trained with LR = 0.15 and LR = 0.001 and weight decay parameter $\lambda = 0.0$ or $\lambda = 0.01$. The total number of epochs is 1000 in all experiments in Fig.11 and all the LR are decayed at epoch 500 to 0.0001. For the experiments in Fig. 15 we use scale-invariant ResNet-18 and project the SGD iterates on the unitary sphere. We test the following LRs in the large-LR phase $(0.0001, 0.0005, 0.00075, 0.001, 0.002, 0.003, 0.004, 0.005)$ to show different generalization performance. After 100 epochs all the learning rates are decayed to the same value 0.0001. In Fig. 15 we fine-tune every 2 epochs for 100 additional epochs with LR=0.0001. To measure the Norm of the Jacobian or the Trace of the Hessian we use a subset of 5000 training datapoints. Each run requires approximately 2 GPU hours on an Nvidia A100 GPU.

**Tiny-ImageNet experiments.** We train Resnet-18 without data augmentation. We use standard SGD *without momentum* in all our experiments. We use the following learning rates $(0.0005, 0.0010, 0.0050, 0.0100, 0.0500, 0.1000, 0.1500, 0.2000, 0.2500)$ and weight decay parameter $(0.0200, 0.0150, 0.0125, 0.0100, 0.0075, 0.0050, 0.0025, 0.0010, 0.0005, 0.0000)$. To measure

the norm of the Jacobian we use a subset of the training data of 2500 examples. Each run requires approximately 5GPU hour on A100.

**LLM experiments.** We use the `NanoGPT` repository (Karpathy, 2023) for training GPT-2 models (Radford et al., 2019) on OpenWebText (Gokaslan et al., 2019). All training documents are concatenated in a single stream from which a new batch is sampled with replacement on every iteration of training. We train a 124M parameter model known as GPT-2-small for $50\,000$ iterations instead of the default $600\,000$ to make grid searches over the learning rate and weight decay parameters more accessible within an academic budget. We mostly use the context length of 256 for faster experiments except for Fig. 8 where we use the context length of 1024 since we observed that a larger context length is crucial to observe loss divergences with moderate learning rates (such as 0.001 for Adam). We train with AdamW (Loshchilov & Hutter, 2019) using batch size 256, default LR 0.0006 (unless mentioned otherwise), $\beta_1 = 0.9$, $\beta_2 = 0.95$, a short 400-iteration LR warmup, and $10\times$ cosine LR decay. For the runs with SGD with momentum, we use the learning rate 0.3 and momentum parameter 0.9 using the same LR schedule as for AdamW. We initialize all parameters with the standard deviation equal to 0.02. We keep all other hyperparameters at their default values as in the `NanoGPT` repository. We perform all experiments on A100 Nvidia GPUs that support fast `bfloat16` training. Each training run of GPT-2-small for $50\,000$ iterations takes around 12 hours on a single GPU.

## C    Weight decay for overparametrized deep networks: additional experiments and details

### C.1    A graphical illustration of the fine-tuning phase

Here, we plot an illustrative graphic in Fig. 9 to give an idea of what happens during the fine-tuning phase.

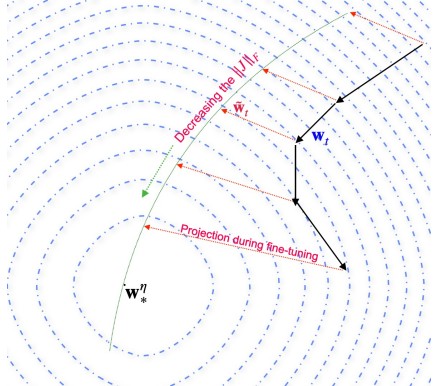

Figure 9: **A graphical illustration of the fine-tuning phase**.

### C.2    Supporting derivations

Here we prove that the scale of noise is well approximated by training loss in the case of binary classification instead of classification in the case of multiple classes. The proof follows the lines of Wojtowytsch (2021).

**Proposition 2.** *Assume* $\|\mathbf{w}\| \in [a, b]$, *for any* $x \in \mathcal{D}$, $\|\nabla h(\mathbf{w}, x)\| \in [m, M]$ *holds. For $n$ sufficiently large, there exists constants $c_1, c_2$ such that*

$$\mathbb{E}\left[\left\|g(\mathbf{w})\right\|^2\right] \leq c_2 \mathcal{L}(\mathbf{w})$$

*Proof.* The noise in the case when the gradient is computed at $(x_i, y_i)$ is

$$g(\mathbf{w}) = \ell'(y_i, h(\mathbf{w}, x_i))\nabla h(\mathbf{w}, x_i) - \frac{1}{n}\sum_i \nabla \ell'(y_i, h(\mathbf{w}, x_i))\nabla h(\mathbf{w}, x_i),$$

Taking the expectation over uniform sampling over $i$, we have,

$$\mathbb{E}\|g\|^2 = \frac{1}{n}\sum_{i=1}^{n}\left(\ell'(y_i, h(\mathbf{w}, x_i))\right)^2\|\nabla h(\mathbf{w}, x_i)\|^2 - \frac{1}{n^2}\Big\|\sum_i \nabla\ell'(y_i, h(\mathbf{w}, x_i))\nabla h(\mathbf{w}, x_i)\Big\|^2 \quad (6)$$

**Upper bound**: Using the self-bounding property of the binary CE, i.e., $\left(\ell'^2\right) \leq l$ and $\left\|\nabla h(\mathbf{w}, x)\right\|^2 \leq M^2$.

$$\mathbb{E}\|g\|^2 \leq M^2\frac{1}{n}\sum_{i=1}^{n}\ell(y_i, h(\mathbf{w}, x_i)) = M^2\mathcal{L}(\mathbf{w}).$$

$\square$

**Comment on the Lower bound**: Since the iterates are bound, we can assume there exists a constant $c$ such that $\left(\ell'^2\right) \geq cl$. as the second term in 6 is decreasing with $O(n^{-2})$, we can assume that the first term is dominating and relevant and can lower bound the first term as,

$$\mathbb{E}\|g\|^2 \geq cm^2\frac{1}{n}\sum_{i=1}^{n}\ell(y_i, h(\mathbf{w}, x_i)) = cm^2\mathcal{L}(\mathbf{w}).$$

### C.3 Additional figures for the over-training regime

In this section, we report additional experimental results related to Section 2 in the main text. In Fig. 10 we report analogous results for the jacobian norm and test error of the EMA for Resnet18 on CIFAR10. In Fig. 11 and 2 we report the train CE for VGG and ResNet18 on CIFAR-10 and ResNet34 trained on CIFAR-100. We can observe how when weight decay is used in combination with large LR, the train CE stabilizes at some approximately constant level.

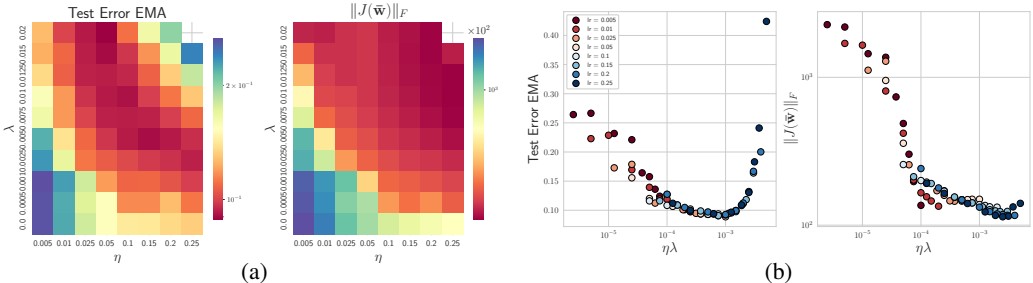

Figure 10: **Resnet18 on CIFAR10** We train Resnet18 on CIFAR10 for 100 epochs with different $\eta$ and $\lambda$. Fig. 10a reports a heatmap of the test error and Jacobian norm for the EMA for all the different combinations of parameters. The test error presents an optimal value of $\eta$ when $\lambda$ is fixed and, consistently with conjecture 1, the Jacobian norm decreases monotonically. More over, Fig. 10b shows how the optimality might depend only on the product $\eta\lambda$ for which the test error has a U-shape and the Jacobian norm decreases monotonically.

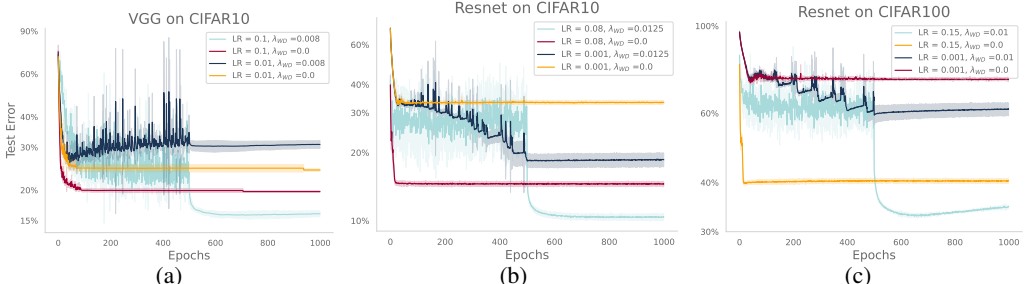

Figure 11: **Training with and w/o weight decay.** We report the test error for VGG (11a) and ResNet (11b, 11c) trained on CIFAR-10/100 with and without weight decay and with small and large learning rates. After the first 500 epochs the learning rate is decayed to $\eta = 10^{-4}$ for all the curves.

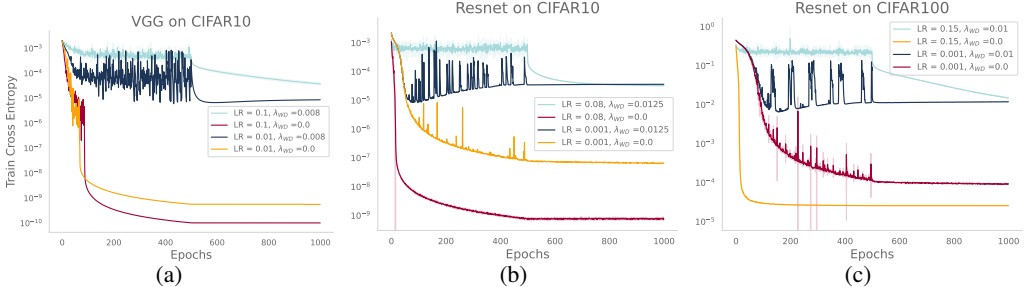

(a)           (b)           (c)

Figure 12: **Training with and w/o weight decay.** We report the train CE for VGG (12a) and ResNet (12b, 12c) trained on CIFAR-10/100 with and without weight decay and with small and large learning rates. After the first 500 epochs the learning rate is decayed to $\eta = 10^{-4}$ for all the curves.

**Connection between SGD covariance and Hessian.** Much of the literature related to implicit bias relies on the assumption that the covariance of the noise of SGD is strictly related to the hessian of the loss function as discussed in Sec 2. Denoting the Hessian $H(\mathbf{w}) := \nabla^2 \mathcal{L}(\mathbf{w})$ we can write it as the so-called Gauss-Newton decomposition (Sagun et al., 2017; Papyan, 2018) $H(\mathbf{w}) = G(\mathbf{w}) + E(\mathbf{w})$. To measure the cosine similarity (CS) between $H(\mathbf{w})$ and the covariance $\Sigma_t$ we compute

$$CS = \mathbb{E}\left[\cos\left(H(\mathbf{w})\boldsymbol{v}, \Sigma_t \boldsymbol{v}\right)\right]$$

where $v$ is sampled from the Gaussian distribution in $\mathbb{R}^p$ and $\cos(\boldsymbol{u}, \boldsymbol{v}) = \langle \boldsymbol{u}, \boldsymbol{v}\rangle / \|\boldsymbol{u}\|\|\boldsymbol{v}\|$. The results are reported in Fig. 13.

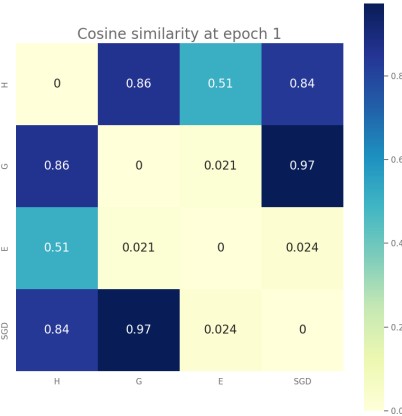

Figure 13: **Cosine similarity between hessian and Noise covariance:** we compute the cosine similarity between the hessian and the covariance of the SGD noise for a scale-invariant ResNet after one epoch with large lr $\eta = 0.005$. The results show how the two matrices are correlated and in particular how the SGD noise covariance is highly correlated with $G(\mathbf{w})$.

## C.4 Trace of Hessian and Jacobian Norm

As reported in Table 1 previous works related with label noise, theoretically derived the connection between the trajectory of SGD and a trajectory which regularizes either the trace of the Hessian or the Jacobian norm. The Hessian of a loss function $\mathcal{L}$ can be decomposed as:

$$\nabla^2 \mathcal{L}(\mathbf{w}) = \sum_{i=1}^{N} \left[ \underbrace{\nabla h(x_i; \mathbf{w}) \left[ \nabla_h^2 l(h(x_i; \mathbf{w})) \right] \nabla h(x_i; \mathbf{w})^\top}_{G_i(\mathbf{w})} + \underbrace{\sum_{c=1}^{K} [\nabla_h l(h(x_i; \mathbf{w}))]_c \nabla^2 h(x_i; \mathbf{w})}_{E_i(\mathbf{w})} \right].$$

Many works demonstrated empirically that the $G_i$ is the dominant part of the Hessian decomposition and $\nabla^2 L(\mathbf{w}) \sim \sum_i G_i$ (Sagun et al., 2017). The Jacobian (J) norm instead is defined as:

$$\left\| J(\mathbf{w}) \right\|_F^2 = \frac{1}{N} \sum_{i=1}^{N} \mathrm{Tr} \left( \nabla h_\mathbf{w}(x_i) \nabla h_\mathbf{w}(x_i)^\top \right) , \tag{7}$$

in the case of square loss, $\nabla_h^2 l = I$ where $I$ is the identity matrix. Hence, $\mathrm{Tr}\left(\nabla^2 L(\mathbf{w})\right) \sim \left\| J \right\|_F^2$. The similarity is an exact equality at an interpolating solution since $\nabla_h l(h(x_i; \mathbf{w})) = 0$ and therefore not much ambiguity is left regarding which quantity to study. However, in the case of classification, this fact does not hold. In particular, the trace of $\nabla_h^2 l$ can significantly deviate from the identity matrix and varies depending on the value of the training loss. Consequently, although the two quantities seem closely related even when using the CE, we opt for analyzing $\left\| J \right\|_F$. This choice is motivated by its lack of explicit dependence on the training loss, enabling straightforward comparisons between different solutions. In the following, we report an empirical comparison of the two quantities along the training trajectory. In Fig. (14) we report a comparison between the trace of the Hessian and the Jacobian norm for both the fine-tuned iterates and the EMA. We can observe that for the fine-tuned iterates, both quantities display a decreasing trend; nevertheless, the ranking appears to be inverted to what is expected from previous theoretical works, i.e. larger LRs should lead to a stronger regularization effect. If we compare the EMA instead, we can see that both quantities are still decreasing along the iterations but for the Hessian we don't observe any meaningful ranking for the final solutions whereas the Jacobian norm is lower for higher LRs.

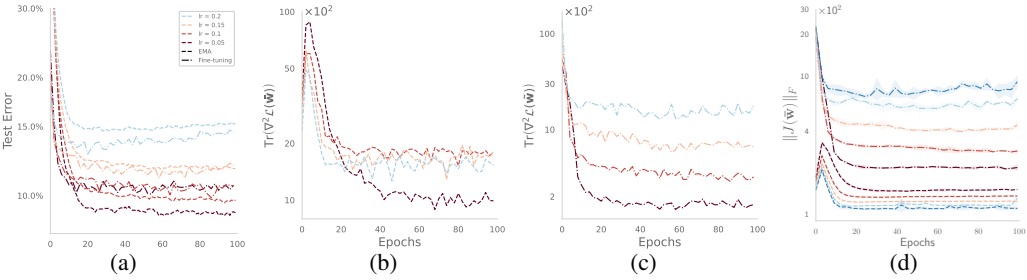

Figure 14: **Trace of Hessian and Jacobian Norm** We train standard Resnet18 on CIFAR-10 for 100 epochs fixing $\lambda = 0.0125$ and varying the learning rate. We report the EMA and we finetune for 100 epochs every 3 with $\eta = 10^{-3}$.

## C.5 Experiments with scale-invariant Resnet on the sphere

In order to isolate the implicit regularization mechanism from the large initial drop and even small fluctuations in the dynamics of the $\ell_2$ norm, we consider a simplified setting. We train scale-invariant networks (Li & Arora, 2019; Li et al., 2020) with projected SGD on the unitary sphere $\mathbb{S}^{(p-1)}$. We project the SGD iterates on the unitary sphere $\mathbb{S}^{(p-1)}$ within the context of scale-invariant ResNet architectures (Li & Arora, 2019; Li et al., 2020). This setup is helpful for two reasons: (a) it simplifies the selection of the LR and hence tremendously reduces the experimental overhead (b) scale-invariant networks have been extensively studied in previous works Li & Arora (2019); Li et al. (2020); Kodryan et al. (2022). Moreover, in the context of scale invariance, the optimization on the sphere is the natural object to study as the evolution of the direction is the only quantity that matters. The

projected SGD update writes as

$$\mathbf{w}_{t+1} = \Pi_{\mathbb{S}^{(p-1)}} \left( \mathbf{w}_t - \eta \nabla_{\mathbf{w}} \ell \left( y_{i_t}, h(\mathbf{w}_t, x_{i_t}) \right) \right) \tag{8}$$

$$\text{where} \quad \Pi_{\mathbb{S}^{(p-1)}} : \mathbf{w} \mapsto \mathbf{w}/\left\| \mathbf{w} \right\|_2. \tag{9}$$

The training framework still consists of two phases separated by a LR decay. The primary insight from our experiments on the sphere is depicted in Fig. 15: the test performance achieved in the fine-tuning phase depends on the LR used in the large-LR phase and, moreover, there is an optimal value. The work of Kodryan et al. (2022) reports a similar observation, where the best test loss is achieved at a LR where the loss neither converges too fast nor diverges but doesn't provide any explanation. Once again, our investigation reveals that the key to understand this behaviour and the dependence on the LR lies in the noisy dynamics in the large LR phase which closely tracks a regularized process. To summarize this idea we postulate a conjecture similar to the one reported in Section 2.2.

**Conjecture 3.** *Consider the algorithm Eq. 8 with $\mathbf{w}_0$ initialized from a distribution $\mu_0 \left( \mathbb{S}^{(p-1)} \right)$. For any input $x$, let $\mathbf{w}_t, h(\mathbf{w}_t, x)$ be the random variables that denote the iterate at time $t$ and its functional value. The stochastic process $(h(\mathbf{w}_t, x))_{t \in \mathbb{N}}$ will converge to a stationary distribution $\mu_\eta^\infty(x)$ with mean $\bar{\mu}_\eta(x)$ for which $\mathbf{w}_\eta^*$ is a first-order stationary point of the following regularized loss:*

$$\bar{\mathcal{L}}(\mathbf{w}) \coloneqq \mathcal{L}(\mathbf{w}) + \eta \sigma_\eta^2 \left\| J(\mathbf{w}) \right\|_F^2. \tag{10}$$

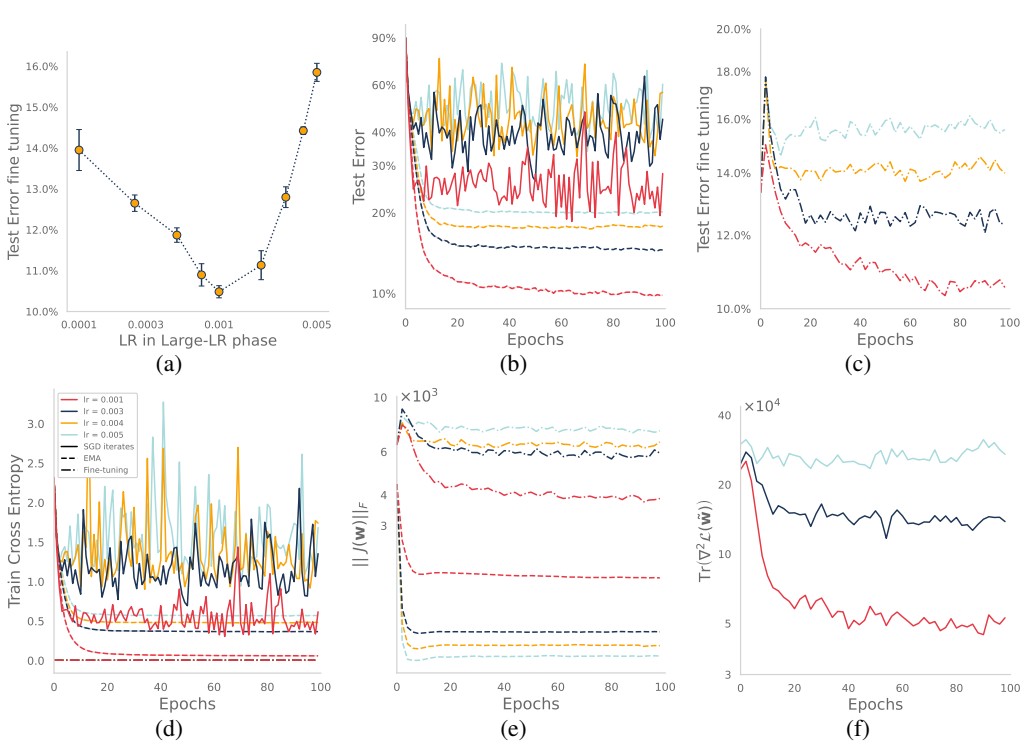

Figure 15: **Training scale-invariant ResNets on the sphere.** We train on CIFAR-10 with different large LR for the first 100 epochs and decay it to $\eta = 10^{-4}$ afterwards. Fig. (15a) reports the test error with respect to different LRs in the first phase showing the existence of an optimal value. Fig. (15b) reports the test error for the SGD iterates ($-$) and for the EMA ($--$). Figure (15c) reports the decreasing trend of the test error after fine-tuning for 100 epochs with $\eta = 10^{-4}$ every 2 epochs. Finally, Fig. (15e) reports the norm of the Jacobian for the EMA and the fine-tuned iterates and Figure (15f) reports a comparison with the trace of the Hessian for the fine-tuning iterates.

## C.6 Empirical validation of Covariance approximation

We empirically verify the validity of the "decoupling approximation" introduced in Sec. 2.2 we use Stochastic Lanczos Quadrature (Yao et al., 2020) to estimate the empirical spectral density of the SGD covariance with and without the decoupling approximation, during the large-LR phase. The experiments are performed for ResNet-18 trained on the cifar10 dataset for different combinations of learning rate and weight decay which are used in the manuscript. The results in Figures 16, 17, and 18 illustrate a substantial overlap in the two spectra which serves as a validation of the reliability of our approximation.

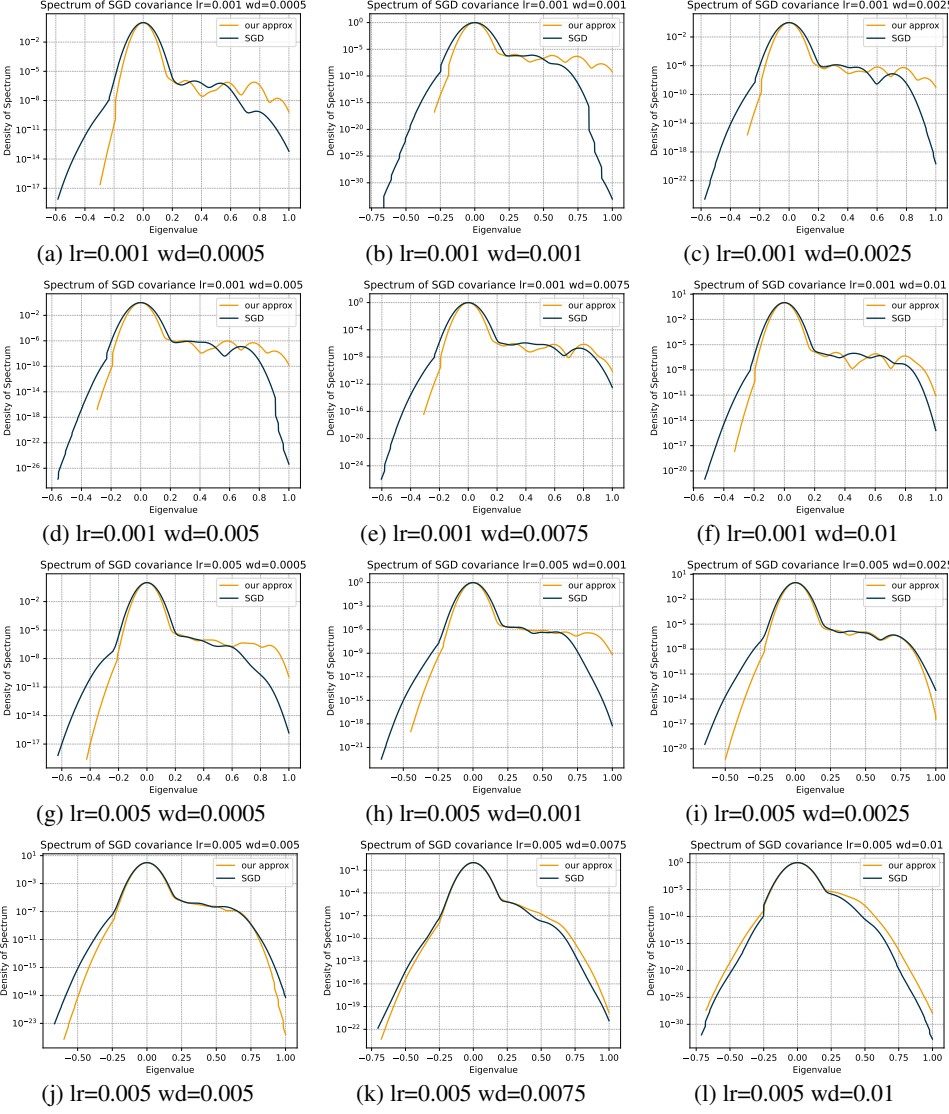

Figure 16: We use Stochastic Lanczos Quadrature to estimate the empirical spectral density of the SGD covariance, with and without the decoupling approximation, during the large-LR phase. Experiments with ResNet-18 on the CIFAR-10 dataset, varying learning rate and weight decay, show a substantial overlap in the two spectra, validating our approximation.

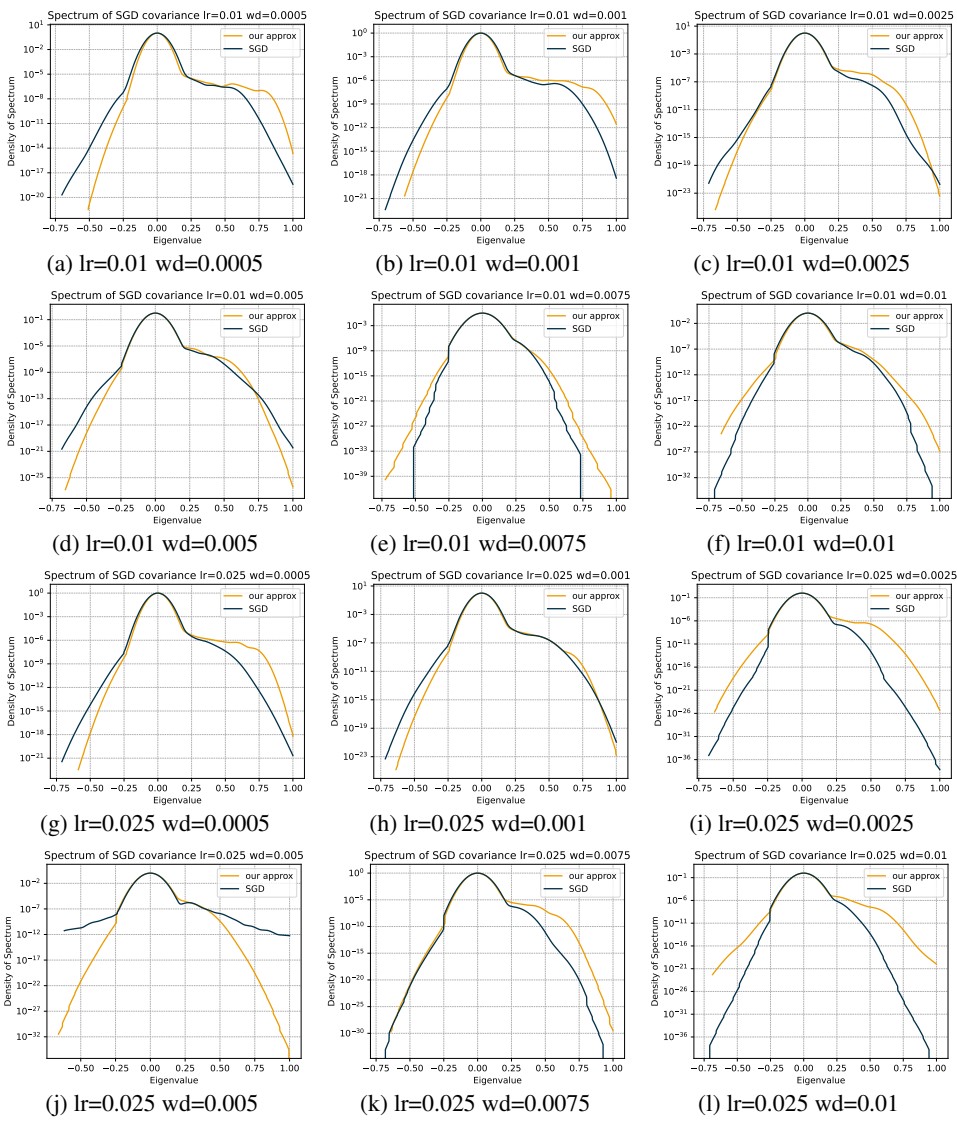

Figure 17: We use Stochastic Lanczos Quadrature to estimate the empirical spectral density of the SGD covariance, with and without the decoupling approximation, during the large-LR phase. Experiments with ResNet-18 on the CIFAR-10 dataset, varying learning rate and weight decay, show a substantial overlap in the two spectra, validating our approximation.

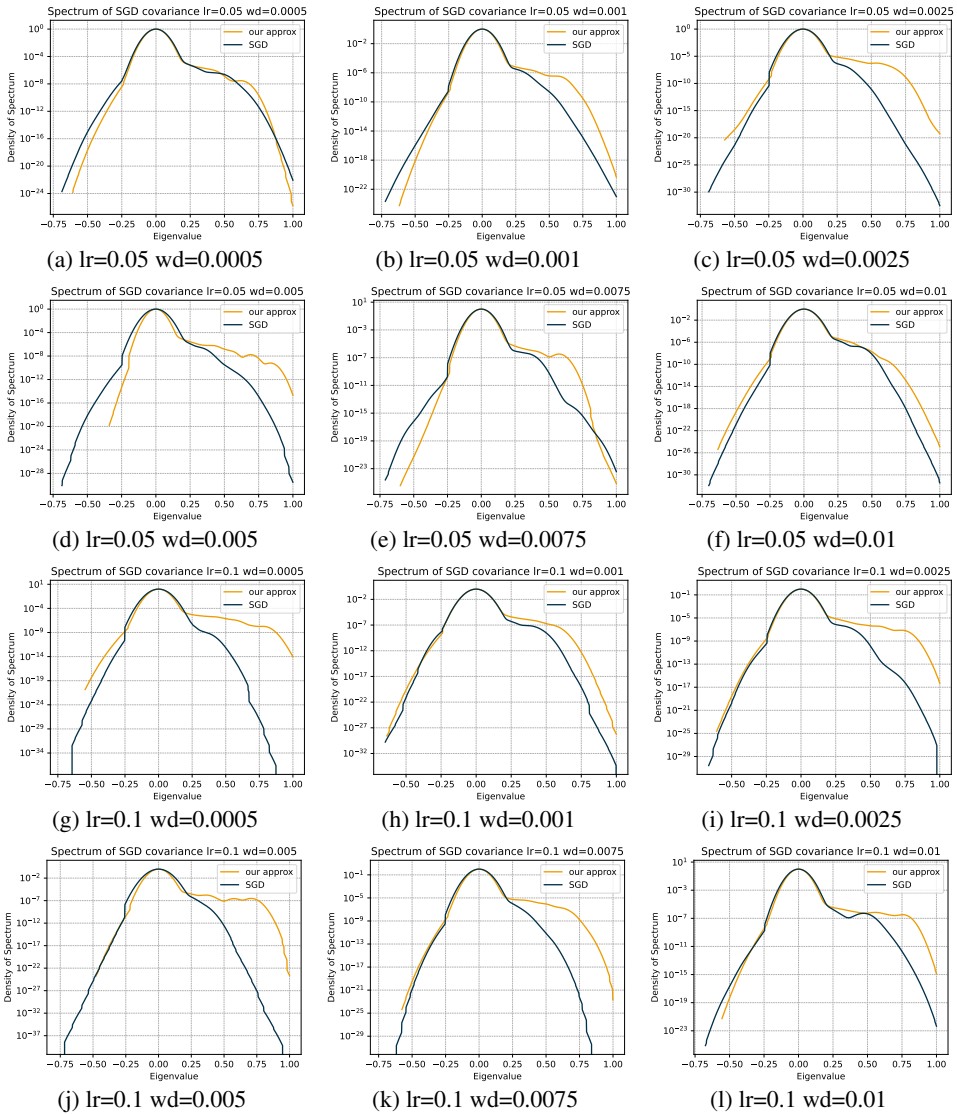

Figure 18: We use Stochastic Lanczos Quadrature to estimate the empirical spectral density of the SGD covariance, with and without the decoupling approximation, during the large-LR phase. Experiments with ResNet-18 on the CIFAR-10 dataset, varying learning rate and weight decay, show a substantial overlap in the two spectra, validating our approximation.

## C.7 Empirical verification of the conjecture through snapshot ensembles

To further validate our conjecture, we performed additional experiments. Specifically, within the same ResNet18 on CIFAR-10 setting as in our main experiments, we created snapshot ensembles (Huang et al., 2017) by averaging in function space along the SGD trajectory every 10 epochs for the combinations of learning rate (LR) and weight decay (WD) considered in the paper.

To assess whether the mean of the stationary distribution in function space aligns closely with the EMA, where the Jacobian norm is regularized, we compared the performance of snapshot ensembles with that of the EMA. Additionally, we computed the Total Variation Distance $\mathcal{D}_{TV}$ between the softmax outputs of the ensemble and the EMA on the Test set

$$\mathcal{D}_{TV} = \frac{1}{2N} \sum_{i=1}^{N} \sum_{j=1}^{C} \left| p_{\text{ensemble},j}^{(i)} - p_{\text{EMA},j}^{(i)} \right|.$$

The results in Table 2 show a strong alignment in test accuracies, while those in Table 3 indicate a low Total Variation across all combinations. Together, these findings offer further validation for our conjecture.

Table 2: Test Error for Snapshot Ensemble and EMA for different values of learning rate (LR) and weight decay (WD).

| WD | LR=0.001 | | LR=0.005 | | LR=0.01 | | LR=0.025 | | LR=0.05 | | LR=0.1 | | LR=0.15 | |
|---|---|---|---|---|---|---|---|---|---|---|---|---|---|---|
| | ENS | EMA | ENS | EMA | ENS | EMA | ENS | EMA | ENS | EMA | ENS | EMA | ENS | EMA |
| 0.0000 | 0.32 | 0.33 | 0.27 | 0.26 | 0.24 | 0.25 | 0.17 | 0.17 | 0.17 | 0.17 | 0.13 | 0.13 | 0.13 | 0.13 |
| 0.0005 | 0.32 | 0.32 | 0.29 | 0.29 | 0.24 | 0.24 | 0.18 | 0.17 | 0.13 | 0.16 | 0.13 | 0.13 | 0.11 | 0.13 |
| 0.0010 | 0.32 | 0.33 | 0.25 | 0.27 | 0.21 | 0.21 | 0.13 | 0.19 | 0.10 | 0.13 | 0.10 | 0.11 | 0.11 | 0.11 |
| 0.0015 | 0.32 | 0.34 | 0.23 | 0.22 | 0.22 | 0.25 | 0.11 | 0.14 | 0.10 | 0.12 | 0.10 | 0.10 | 0.09 | 0.09 |
| 0.0025 | 0.30 | 0.30 | 0.22 | 0.22 | 0.19 | 0.20 | 0.09 | 0.10 | 0.10 | 0.11 | 0.10 | 0.10 | 0.10 | 0.11 |
| 0.0050 | 0.33 | 0.34 | 0.21 | 0.20 | 0.12 | 0.16 | 0.10 | 0.10 | 0.10 | 0.10 | 0.09 | 0.09 | 0.10 | 0.09 |
| 0.0075 | 0.35 | 0.37 | 0.15 | 0.16 | 0.10 | 0.11 | 0.11 | 0.11 | 0.09 | 0.08 | 0.10 | 0.09 | 0.12 | 0.10 |
| 0.0100 | 0.31 | 0.34 | 0.13 | 0.15 | 0.11 | 0.11 | 0.10 | 0.10 | 0.11 | 0.10 | 0.11 | 0.09 | 0.13 | 0.13 |

Table 3: Total Variation Distance between softmax output of Ensemble and EMA.

| WD | LR=0.001 | LR=0.005 | LR=0.01 | LR=0.025 | LR=0.05 | LR=0.1 | LR=0.15 |
|---|---|---|---|---|---|---|---|
| 0.0000 | 0.03 | 0.02 | 0.01 | 0.01 | 0.01 | 0.01 | 0.01 |
| 0.0005 | 0.04 | 0.02 | 0.02 | 0.03 | 0.09 | 0.08 | 0.07 |
| 0.0010 | 0.04 | 0.05 | 0.04 | 0.10 | 0.09 | 0.07 | 0.07 |
| 0.0015 | 0.04 | 0.07 | 0.07 | 0.08 | 0.08 | 0.07 | 0.07 |
| 0.0025 | 0.04 | 0.11 | 0.10 | 0.06 | 0.08 | 0.08 | 0.09 |
| 0.0050 | 0.06 | 0.15 | 0.11 | 0.09 | 0.08 | 0.10 | 0.11 |
| 0.0075 | 0.08 | 0.12 | 0.10 | 0.10 | 0.10 | 0.11 | 0.12 |
| 0.0100 | 0.10 | 0.09 | 0.10 | 0.10 | 0.11 | 0.12 | 0.13 |

## D   Weight decay for large language models: additional figures and details

We present the following additional figures related to the LLM experiments. We show that the validation loss of a GPT-2-124M model is determined by the training loss and not influenced by $\lambda$ in Fig. 19. We also show that the generalization gap stays close to zero throughout training for different $\lambda$ for both 124M and 774M parameter models. We show the results for models trained *weight decay on LayerNorm weights* in Fig. 20. We see that penalizing all parameters in weight decay (i.e., including the LayerNorm parameters) leads to the same effect for smaller $\lambda$ (like 0.1) but underperforms on larger $\lambda$ (like 0.3). Note that when WD is applied on all weights, this changes the optimal value of the objective. In Fig. 21, we train models with $\ell_2$ regularization instead of decoupled weight decay as in AdamW (Loshchilov & Hutter, 2019). We observe that $\ell_2$ regularization instead of weight decay leads to the same effect as decoupled weight decay (Loshchilov & Hutter, 2019). We train models using SGD with momentum and show the results in Fig. 22. We see that weight decay leads to a similar improvement in training loss for SGD with momentum as well. We show multiple metrics in Fig. 26 for the models shown in Fig. 6: gradient variance, gradient norm, and weight norm plots that complement Fig. 7 in the main part. In Fig. 23, we show results of weight averaging that suggests the suboptimality gap between runs with different $\lambda$ is much smaller than what the loss at $w_t$ suggests. However, weight averaging is still less effective than fine-tuning with a tiny LR as in Fig. 6. Finally, in Fig. 25, we show results of models trained context length 1024. We see that the training loss over iterations for models trained with a range of LR and WD (all are `bfloat16`). All runs with LR smaller than 0.001 successfully converge but the final training loss is higher than for LR 0.001. In addition, we observe that lower learning rates prevent the weights from growing too much.

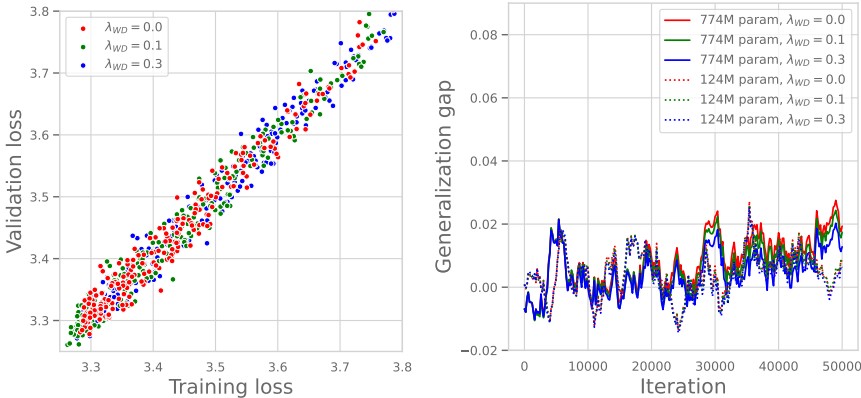

Figure 19: **Left**: The validation loss of a GPT-2-124M model is determined by the training loss and not influenced by $\lambda$. **Right**: The generalization gap stays close to zero throughout training for different $\lambda$ for both 124M and 774M parameter models.

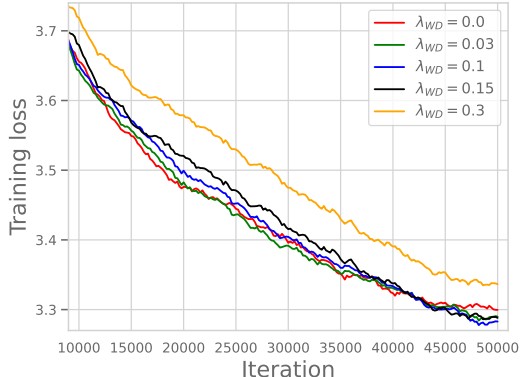

Figure 20: **GPT-2-124M on OpenWebText with weight decay on LayerNorm weights.** Penalizing all parameters in weight decay (i.e., including the LayerNorm parameters) leads to the same effect for smaller $\lambda$ (like 0.1) but underperforms on larger $\lambda$ (like 0.3). Note that when WD is applied on all weights, this changes the optimal value of the objective.

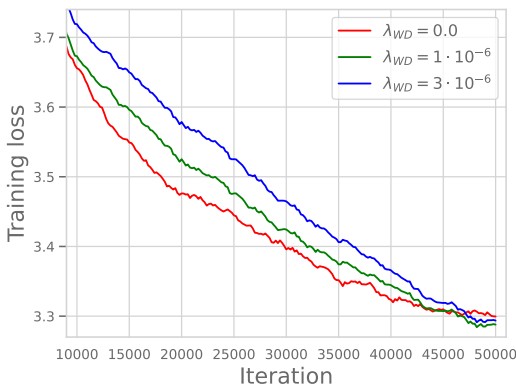

Figure 21: **GPT-2-124M on OpenWebText with $\ell_2$ regularization.** We observe that $\ell_2$ regularization instead of weight decay leads to the same effect as decoupled weight decay (Loshchilov & Hutter, 2019).

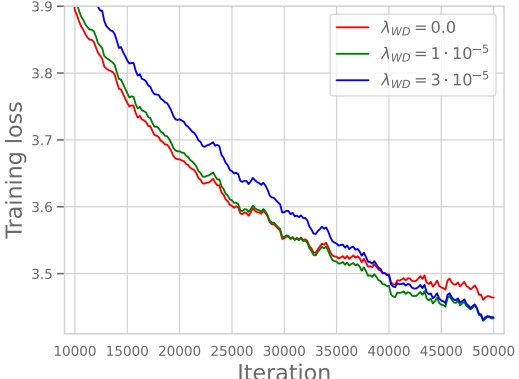

Figure 22: **GPT-2-124M on OpenWebText trained with SGD with momentum.** Weight decay leads to a similar improvement in training loss for *SGD with momentum* as well (all other experiments are done with AdamW).

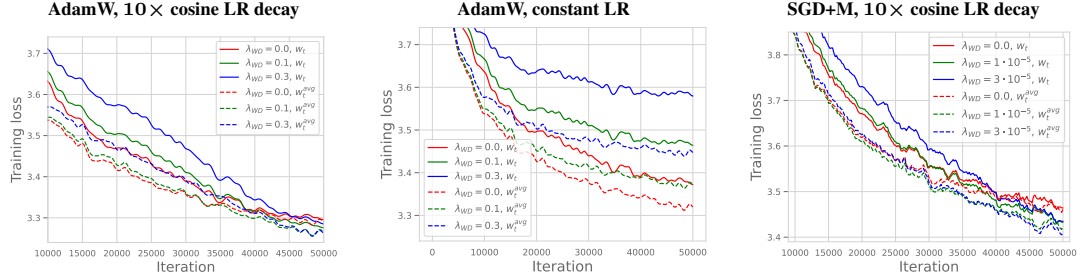

Figure 23: **Weight averaging for GPT-2-124M on OpenWebText.** Weight averaging ($w_t^{avg}$) shows that the suboptimality gap between runs with different $\lambda$ is much smaller than what the loss at $w_t$ suggests. However, weight averaging is still less effective than fine-tuning with a tiny LR as in Fig. 6.

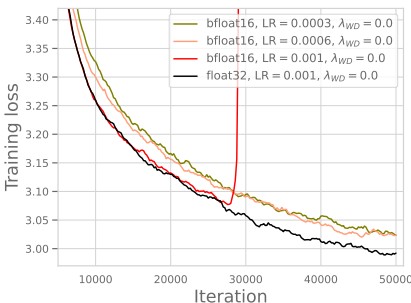

Figure 24: **GPT-2-124M on OpenWebText with context length 1024.** The model trained with a moderate LR 0.001 diverges for `bfloat16` but not for `float32`; lowering the LR prevents the divergence but leads to a worse loss.

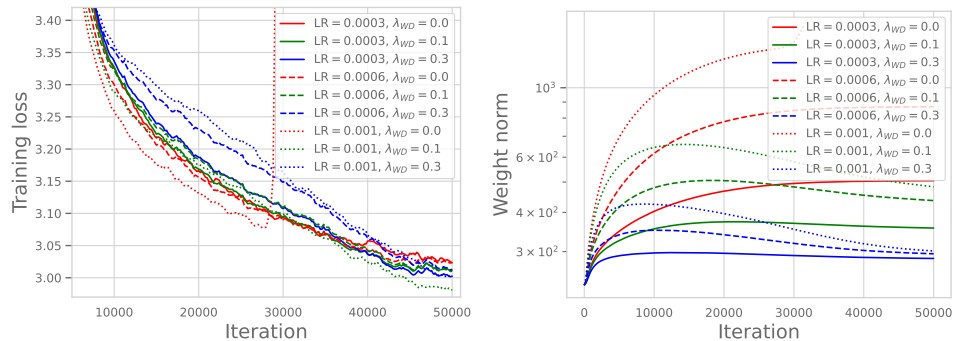

Figure 25: **GPT-2-124M on OpenWebText with context length 1024.** (*Left*) The training loss over iterations for models trained with a range of LR and WD (all are `bfloat16`). All runs with LR smaller than 0.001 successfully converge but the final training loss is higher than for LR 0.001. (*Right*) Weight norms for LR in $0.0003, 0.0006, 0.001$ for $\lambda = 0.1$ which does not diverge. Lower learning rates prevent the weights from growing too much.

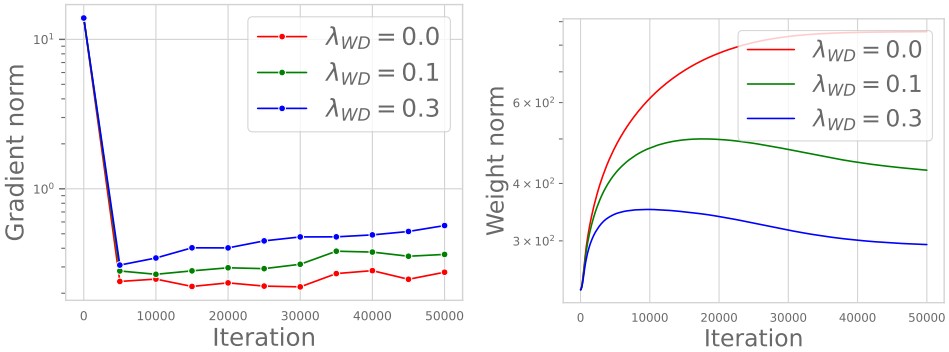

Figure 26: **GPT-2-124M on OpenWebText.** The gradient norm and weight norm plots for the models reported in Fig. 7.

