# OpenReview forum: "Why Do We Need Weight Decay in Modern Deep Learning?"
_NeurIPS.cc/2024/Conference — NeurIPS 2024 poster_

### Official Review · Reviewer_tBnp · 2024-07-06

**Soundness:** 2
**Presentation:** 3
**Contribution:** 2
**Rating:** 5
**Confidence:** 4

**Summary:**

In this work, the role of weight decay (WD) in contemporary deep learning is investigated. The authors note that interpreting weight decay as a classical $L_2$ regularizer seems less relevant for modern deep learning. Instead, it is suggested to separately regard WD from positions of over- and under-training. In the over-training regime, typical for common image classification with ResNets on CIFAR-like datasets, weight decay paired with learning rate acts as an implicit regularizer on the Jacobian of the model, controlling the scale of the noise defined by the derivative of the loss. Optimal combination of weight decay and learning rate thus yields the best generalization due to a balanced regularization. In the under-training regime, attributed to a single-pass LLMs training, WD takes a different role controlling the effective learning rate of the training process and saving it from numerical instabilities.

**Strengths:**

Despite a variety of literature examining the widely used weight decay technique, there still lacks precise understanding of its mechanisms in the context of deep neural networks training. The idea to consider its role from the dual perspective of over- and under-training regimes is interesting and indeed relevant for modern deep learning. The paper is well written and structured accordingly, its claims are stated clearly and justified by multiple experiments. The mutual role of weight decay and learning rate acting as an implicit regularizer is intriguing while intuitive, especially in light of prior work studying the effective learning rate in scale-invariant architectures and promoting the significance of large step sizes for better generalization. It is remarkable that in the context of LLM training, WD provides other benefits: it implicitly controls the pace of optimization, as shown both theoretically by examining a simplified Adam update and experimentally, and empirically prevents training instabilities.

**Weaknesses:**

**Over-training regime**

My personal expertise is the over-training regime, hence most questions and weaknesses are attributed to the first part of the work.
1. The mechanism, by which WD helps in the over-training regime via constraining the optimization on a sphere, as described in lines 137-138, is unconvincing. Would similar reasoning be applicable to other loss functions not requiring infinite increase of parameters norms, such as MSE, or fully scale-invariant models, which also benefit from the use of WD during training but do not depend on the parameters norm by definition?
2. Equation 3 involves a series of unrealistic approximations. First of all, where did the total loss gradient go after the first approximation? According to the previous assumption, the loss is separated from its minimum value by WD, thus its gradient must be substantially non-zero. Second, the assumption that the first derivative is approximately constant across all data points is too stringent as the loss on individual objects has a heavy-tailed distribution in practice, where the mean and median values may be very distinct.
3. The Gaussian approximation of the SGD noise is questionable [1-3] and must be justified experimentally or at least by citing related work.
4. Conjecture 1 is formulated in a concrete form, however only proxy values are measured to justify it. (see Question 3 below for further detail).
5. It is unclear how regularization of model Jacobian and generalization are related in the first place, as trends plotted in Fig. 3 and 4 could be mere correlations caused by other intrinsic factors [4]. In particular, in Figure 5, the authors note that the Jacobian norm trends for EMA and fine-tuning are inverse, making it hard to conclude whether the Jacobian norm is regularized or actually fostered during the large-LR stage of training.
6. Dependence of the training dynamics on the product of learning rate and weight decay is not novel and was particularly studied, for example, in [5] where this value is called intrinsic learning rate.

**Under-training regime**

Overall, I like this part of the work more, especially the derivation of the effective learning rate for Adam and its empirical examination. However, I still have a few concerns.

7. The informal explanation of the AdamW training dynamics using bias-variance reasoning is vary vague and not supported by any experiments. I suppose that similar linear least-squares arguments could be similarly applied to the over-training regime, where WD is expected to play a different role according to the claims of the paper.
8. Although empirically validated, no explanation is offered as to how the use of WD allows avoiding numerical instabilities in LLM training.

[1] Simsekli, Umut, Levent Sagun, and Mert Gurbuzbalaban. "A tail-index analysis of stochastic gradient noise in deep neural networks." International Conference on Machine Learning. PMLR, 2019.

[2] Panigrahi, Abhishek, et al. "Non-Gaussianity of stochastic gradient noise." arXiv preprint arXiv:1910.09626 (2019).

[3] Zhou, Pan, et al. "Towards theoretically understanding why sgd generalizes better than adam in deep learning." Advances in Neural Information Processing Systems 33 (2020): 21285-21296.

[4] Andriushchenko, Maksym, et al. "A Modern Look at the Relationship between Sharpness and Generalization." International Conference on Machine Learning. PMLR, 2023.

[5] Li, Zhiyuan, Kaifeng Lyu, and Sanjeev Arora. "Reconciling modern deep learning with traditional optimization analyses: The intrinsic learning rate." Advances in Neural Information Processing Systems 33 (2020): 14544-14555.

**Questions:**

1. Figure 2: How come the huge spikes of the dark blue line in the training loss and test error are absolutely not reflected in the weight norm?
2. Equation 4: How exactly does this formula approximate the SGD update in eq. (2)? Where is the WD factor $(1 - \eta \lambda)$ before $w_t$? Why is a single $g_t$ term replaced with a sum of normally distributed values?
3. According to the formulation of Conjecture 1, it seems more natural to perform MCMC ensembling of the models in the training trajectory to approximate the mean value $\bar{\mu}_{\eta, \lambda} (x)$ rather than doing EMA or fine-tuning (the authors discuss limitations of these methods in Sec. 2.3). I would suggest to conduct such experiment to more accurately validate the conjecture.
4. Does Fig. 4 also report the EMA results?
5. How does the authors derivation of the effective learning rate for Adam compare with a similar prior result obtained for Adam in the scale-invariant case [1]? I acknowledge that the authors result could be more general, albeit it still requires independence of the gradient sign from the parameters scale akin to the scale-invariant case.

[1] Roburin, Simon, et al. "Spherical perspective on learning with normalization layers." Neurocomputing 487 (2022): 66-74.

**Limitations:**

The authors partly discuss the limitations in the conclusion of the work.

---

> ### Author Rebuttal · Authors · 2024-08-07
>
> We thank the reviewer for the detailed comments.
>
> 1\. The main focus of our work is classification because, at the core of our examination, are the unique properties of exponentially tailed loss functions, such as cross-entropy (CE), as explained in Section 2.1. Therefore, our reasoning does not extend to MSE. Regarding scale-invariant models, [1] shows that training with SGD and WD also leads to the norm stabilizing. Additionally, in Appendix C.6, we investigate training scale-invariant models on the sphere as a proxy for WD. Our empirical results show analogous phenomena in this setting: stabilization of the loss function, a decrease in the Jacobian norm with a correspondent improvement in test error.
>
> 2\. See points (1,2,3) in the main response.
>
> 3\. See (4) in the main response.
>
> 4\. The conjecture has been formulated for clarity of exposition and summarize the main message of this part of the paper.
>
> 5\. Previous works ([6], [7], [8]) already show that explicitly regularizing the Hessian or Jacobian of the network is beneficial for generalization. Regarding the discrepancy between fine-tuning and EMA, we would like to point out that in both cases, the Jacobian norm decreases during the large LR phase and is never fostered. The difference arises with different LRs, where the final Jacobian norm is expected to be smaller for larger LR, as seen with EMA. As explained in Section 2.3, fine-tuning is not ideal for studying dynamics with large LR because it may deviate from the mean process, potentially optimizing a different objective. Therefore, we believe that the results with fine-tuning do not compromise the validity of our conclusions regarding the mean process and EMA.
>
> 6\. We agree that the role of the product of LR and WD is not novel, nevertheless in [1] this product is not related to generalization but only to the effective speed of learning and to the equilibrium distribution. Therefore, the mechanism through which different intrinsic LRs affect the generalization of the correspondent equilibrium distribution remains an open question which we instead try to address. Moreover, the work heavily relies on having fully scale-invariant models, which is not needed in our framework therefore making it more general.
>
> 7\. Our bias-variance decomposition applies only to single-pass SGD, as multi-pass SGD lacks sample independence, invalidating the decomposition in the overtraining regime. We verify the bias-variance tradeoff through fine-tuning (FT) at various iterations (Fig. 6), identifying phases where bias or variance dominates. Early FT, where variance is high, reduces the variance revealing a sufficient contraction of the bias, aiding later phases. In contrast, late FT has minimal impact as the variance is already low. This tradeoff also explains the peculiar loss profiles with high ELR, showing higher initial loss and lower final loss (lines 341-346).
>
> 8\. We suspect that the numerical instability comes from `bfloat16`'s limited precision, which can occur when different components in the network with varying scales are added together (e.g., values in the residual stream). Since the `bfloat16` data type has the same exponent size as `float32`, high-weight norm alone should not pose a problem. However, the precision of `bfloat16` is very limited, e.g., `bfloat16(256.0) + bfloat16(1.0)` is equal to `256.0` instead of `257.0`. We suspect this precision limitation is the primary problem in `bfloat16` runs without WD.
>
> ## Questions:
> 1. The spikes in the training loss do not necessarily need to be reflected in the L2 norm of the weights. These spikes can occur due to the trajectory passing through sharp regions of the landscape, where the loss value changes significantly even with small changes in the weights. Consequently, the L2 norm may remain relatively stable while the loss exhibits large fluctuations.
>
> 2. We thank the reviewer for pointing out the missing factor $(1-\eta \lambda)$. The fact that $g_t$ can be written as a sum of random vectors stems from our Gaussian approximation and the structure of SGD's covariance, where its noise is spanned by the gradients [9]. We can consider the random vector $z_t := \frac{1}{\sqrt{N}} \sigma_{\eta,\lambda}(w_t) \sum_{i = 1}^{N} \nabla h_{w_t}(x_i) \, \xi_i^t$  with  $\xi_i^t \sim \mathcal{N}(0,1)$ and verify that $z \sim \mathcal{N}(0,\Sigma_{w_t})$. First of all it is clear that $\mathbb{E}[z_t] = 0$, furthermore we can compute the covariance by  $\mathbb{E}[z_t z_t^\top] = \frac{1}{N} \sigma^2_{\eta,\lambda}(w_t) \sum_{i = 1}^{N} \sum_{j = 1}^{N} \nabla h_{w_t}(x_i) \,  \nabla h_{w_t}(x_j)^\top \mathbb{E}[  \xi_i^t \xi_j^t ]$   the latter double summation is non zero only when $i=j$ therefore  we recover the covariance of SGD.
>
> 3. Sampling from the stationary distribution of SGD would be interesting, but we lack the analytical form of this distribution, making MCMC sampling inapplicable. We argue that the SGD trajectory at stationarity corresponds to samples from a stationary distribution, whose mean is a stationary point of the regularized loss in Equation 5. Although we do not provide an analytical expression for it, one approach could be MCMC sampling from the Gibbs measure using the Jacobian-regularized loss as an energy function. However, there is no guarantee that this would be the stationary distribution of SGD, and this method is computationally demanding due to the prohibitive cost of calculating the Jacobian for the entire dataset.
>
> 4. Yes, Fig. 4 also reports EMA results. Notice the axis labels ("Test Error EMA"), we will clarify in the caption.
>
> 5. We derive the effective LR of Adam by leveraging its (asymptotic) similarity with SignGD, unlike [16] which directly derived it for Adam. This leads to a *simpler* estimate of the ELR, which shows excellent empirical agreement (Fig. 7). Our estimate still requires independence from the scale, which applies with scale-invariance or when the norm remains approximately constant

---

> > ### Comment · Reviewer_tBnp · 2024-08-10
> > **Reviewer's response**
> >
> > Thank you for the detailed discussion and additional experiments!
> >
> > > Sampling from the stationary distribution of SGD would be interesting, but we lack the analytical form of this distribution, making MCMC sampling inapplicable. We argue that the SGD trajectory at stationarity corresponds to samples from a stationary distribution, whose mean is a stationary point of the regularized loss in Equation 5.
> >
> > We do not need the analytical form of the distribution, if we reached stationarity. In that case, simple SGD iterates would behave similarly to the Langevin dynamics on the desired distribution. I would suggest conducting the following experiment: after SGD reached stationarity (according to some proxy metrics, e.g. training loss stabilization), try to ensemble consequent iterates, as is done in Snapshot Ensembles. The ensembled model would closely approximate the mean value of the distribution in the function space, if we truly reached stationarity.
> > I wonder, what is the discrepancy between such a model and the EMA/fine-tuned network? If all three are somewhat similar, then the claims made in the paper would be much better supported, to my view.
> >
> > ---
> >
> > After reading the author's rebuttal and other reviews, I would like to increase my score by one point.
> >
> > To recommend acceptance, I would expect better justification of theoretical approximations and more clear evidence supporting the specific formulation of the main conjecture made in the paper.
> > The cited literature is legit, however, I wanted to see more evidence related to the specific setup considered in this work, e.g. I still doubt that training with sufficiently large LR values for the SGD dynamics to stabilize at some constant loss level would ensure that the total loss gradient is negligible (compared to a near-converged training state considered in prior works). Also, there seem to be as many papers supporting the Gaussian approximation as there are against it (I provided a few references in the review), therefore I was interested why this approximation is relevant for this particular study with its specifics.

---

> > > ### Author Response · Authors · 2024-08-13
> > > **Additional experiments with Snapshot Ensembles**
> > >
> > > We appreciate your suggestion for the additional experiment and your consideration of increasing your score.  We conducted the experiment as you suggested, and the results have been posted as a general comment above. We hope these new findings help to further clarify the doubts you raised.

---

> > > > ### Comment · Reviewer_tBnp · 2024-08-13
> > > > **Reviewer's response to additional experiments**
> > > >
> > > > Thank you for experimentally validating my idea!
> > > >
> > > > I find the results quite interesting and convincing. I would like to increase my score be one more point and suggest the authors including these results in the next revision, as they significantly strengthen the claims of the paper.

---

### Official Review · Reviewer_g6ok · 2024-07-12

**Soundness:** 3
**Presentation:** 3
**Contribution:** 2
**Rating:** 6
**Confidence:** 3

**Summary:**

This paper investigates the role of weight decay (WD) in modern deep learning, differentiating its effects in over-training and under-training regimes. Through experiments with ResNets on vision tasks (over-training) and Transformers on text data (under-training), the authors demonstrate that WD’s primary benefit is not explicit regularization, but rather its influence on optimization dynamics.  In the over-training regime, WD with large learning rates leads to non-vanishing SGD noise, controlling the Jacobian norm and improving generalization. In the under-training regime, WD functions as a modified effective learning rate, improving training loss and stability, particularly with bfloat16 precision.

**Strengths:**

* The paper provides a valuable review of prior work investigating the role of weight decay in modern neural networks, highlighting the need for a deeper understanding of its impact on optimization dynamics.
* A key strength is the clear distinction made between over-training and under-training regimes,  demonstrating that WD's function can vary drastically depending on the training context.
* The paper effectively alternates between theoretical analysis (despite its highly approximate nature) and empirical experiments. Notably, the presented experiments are well-designed to confirm predictions derived from the paper's conjectures and hypotheses, demonstrating a strong consistency between predictions and observations.

**Weaknesses:**

* Although the paper provides a compelling narrative connecting WD's influence to optimization dynamics, it doesn't offer concrete, actionable guidelines for practitioners on how to tune WD for different model architectures or datasets. Insights partially boil down to "instead of weight decay, other regularizer that avoid loss-collapse would have a similar effect".
* The theoretical analysis, while providing some interesting insights, rely heavily on approximations and lacks formal mathematical proofs for the proposed conjectures. This leaves room for inaccuracies not covered by the simplified theoretical framework.
* Especially the language experiments in the under-training regime use tiny models by today's standard. One can indeed hope though, that the results generalize to bigger architectures.

**Questions:**

From Figures 3 and 10, but also my own experience: A meaningful improvement in test-performance often requires carefully tuned WD, within at most a factor of 2. Would you agree with that observation? Many of the experiments show ablations with significantly larger differences in WD strength. Is that an issue for any of the conclusions described in the paper?

**Limitations:**

See above.

---

> ### Author Rebuttal · Authors · 2024-08-07
>
> We thank the reviewer for their helpful comments. Below, we address the concerns raised by the reviewer in detail
>
> >Although the paper provides a compelling narrative connecting WD's influence to optimization dynamics, it doesn't offer concrete, actionable guidelines for practitioners on how to tune WD for different model architectures or datasets. Insights partially boil down to "instead of weight decay, another regularizer that avoid loss-collapse would have a similar effect".
>
> **Practical takeaways:** While our paper primarily focuses on providing a conceptual understanding of how weight decay influences optimization dynamics and improves generalization, we believe our findings also offer practical takeaways. In practice, both weight decay and learning rate are often tuned through extensive grid searches considering combinations of both parameters. However, our results in Figures 3 and 4 demonstrate that the key parameter is actually the product of the two, $\eta \times \lambda$. This insight suggests that a 2D grid search is not necessary. The existence of a connected region of good performance where the product is constant implies that practitioners can simplify their tuning process. I.e., it is not needed to tune both at the same time but it is sufficient to fix $\eta$ and find the best $\lambda$ or vice-versa.
>
>
> >The theoretical analysis, while providing some interesting insights, rely heavily on approximations and lacks formal mathematical proofs for the proposed conjectures. This leaves room for inaccuracies not covered by the simplified theoretical framework.
>
> **Conjecture and approximations:** We acknowledge that our results regarding implicit regularization are presented as conjectures. This decision was made consciously, as we believe it is challenging to rigorously prove these results. However, we included these conjectures to summarize our intuition and facilitate the understanding of our main message, providing a conceptual framework that could guide future research. Regarding the approximations in our analysis, we would like to clarify a few points. First, our series of approximations do not need to hold throughout the entire trajectory of SGD, but only during the stabilization phase. This is an important detail that we will clarify in the manuscript. Additionally, we will better justify our approximations in the revised manuscript in the context of the related literature. In particular, the work of [1] demonstrated that modelling the SGD noise by a Gaussian is sufficient to understand its generalization properties. Moreover, assuming the first derivative constant across datapoints was introduced and empirically verified in [2] (Figure 1, page 10). To further verify the latter assumption in our setting, we performed additional experiments comparing the spectrum of the SGD covariance with that of our approximation at stabilization. The results illustrate a substantial overlap in the spectrum, as detailed in the supplementary material (see the attached PDF in the main response).
>
> > Especially the language experiments in the under-training regime use tiny models by today's standard. One can indeed hope though, that the results generalize to bigger architectures.
>
> **Scaling challenges:** We agree that GPT-2-scale models are considered tiny now, and we acknowledge this limitation of our work. It is obviously challenging for a typical research lab to train significantly larger language models, especially when also having to perform grid searches over multiple hyperparameters.
>
> > From Figures 3 and 10, but also my own experience: A meaningful improvement in test performance often requires carefully tuned WD, within at most a factor of 2. Would you agree with that observation? Many of the experiments show ablations with significantly larger differences in WD strength. Is that an issue for any of the conclusions described in the paper?
>
> **Question regarding tuning weight decay:** In our preliminary experiments, we explored an even finer grid for weight decay values than those reported in Figure 3. However, we did not observe any substantial differences in test performance for these finer values, at least in the overtraining regime. The only difference would have been a higher resolution for the heatmap that would lead to the same conclusions. Furthermore, as shown in Figure 3, despite the larger differences in $\lambda$ as highlighted by the reviewer, we found that similar test performances can be achieved with proper tuning of the LR. This aligns with our main practical takeaway: one can fix one of the parameters (either weight decay or LR) and tune only the other to achieve optimal performances.

---

> > ### Comment · Reviewer_g6ok · 2024-08-12
> >
> > Thank you. I have read the other reviews and your replies carefully. Like some of the other reviewers, I too consider the analysis and discussion of the over-parametereized regime the more interesting and novel one in this work.
> >
> > I believe this work contributes to the already vast body of literature on weight decay -- even though maybe not all aspects the the analysis will withstand the test of time in the long run. But your rebuttal and the additional experiments have strengthened the paper, which is interesting and well executed.
> >
> >
> > Overall, I maintain my score and hope the paper will be accepted.

---

### Official Review · Reviewer_S7kS · 2024-07-13

**Soundness:** 2
**Presentation:** 3
**Contribution:** 3
**Rating:** 5
**Confidence:** 4

**Summary:**

In the paper the authors investigate WD in current deep learning practices. The authors argue that for deep networks used in vision tasks, WD enhances implicit regularisation by modifying the optimisation through loss stabilisation. They also argue that for LLMs, WD helps balance the bias-variance trade-off, leading to improved training stability. The study offers a unified perspective on the role of WD across different training regimes, suggesting that WD’s benefits are more related to changes in training dynamics than to explicit regularization effects.

**Strengths:**

The paper has a number of strengths:
- It presents new insights into how it impacts optimisation dynamics rather than merely acting as a regulariser
- It offers practical recommendations for tuning WD and learning rates, providing actionable insights for practitioners
- The authors set out a theoretical basis for the empirical observations, linking them to the underlying optimization principles

**Weaknesses:**

The weaknesses are as follows:
- The study primarily focuses on ResNets and GPT-2. It would have been interesting to see how the results scale to larger models, such as the open source Llama-3. This fact makes it harder to generalise the results.
- No comparison with other regularisation methods, like batch norm or dropout. I'm not saying this is necessary for the paper to be accepted but it would be a considerable added strength to have at least some of the experiments on other regularising techniques for comparison
- The study assumes fixed hyperparameters in many experiments, which may not reflect the dynamic tuning often required in real-world training
- Lack of justification and explanation around assumptions (see questions)

Minor comments:
- Font in figures is far too small and thin, making it very hard to read
- The use of "weight decay" vs "WD" in the main text is mixed and inconsistent

**Questions:**

- There is an assumption that the first derivative is approximately constant across all datapoints (eq 3), what is this assumption based on? Will this hold across all types of data and data complexity, including multimodal data?
- Related to the previous question, does the data have to be homogenous?
- In the sections discussing the benefits of weight decay with bfloat16 precision, could the authors elaborate on the mechanisms through which weight decay interacts with lower precision calculations to enhance stability?

**Limitations:**

The discussion on limitations is severely lacking.

There is one mention of a limitation of computational resources which hinder large scale experiments but I cannot find any other mention of limitations. This is not sufficient. The authors make a number of assumptions in their experimental setups and in the interpretation of the findings (I mention only a few in my weakness list). The authors must provide a more thorough discussion of the limitations of their work.

---

> ### Author Rebuttal · Authors · 2024-08-07
>
> We thank the reviewer for their comments. We acknowledge the lack of a detailed discussion of the limitations of our work. We did mention the main limitations in the conclusions (no truly large-scale experiments, no proofs of new theoretical results), we will improve upon this by adding a paragraph discussing the limitations due to our approximations. We also elaborate on them below.
>
> **Assumptions and approximations in SGD:** We would like to clarify that our series of approximations does not need to hold along the entire trajectory of SGD but only in the stabilization phase. This is an important detail that we will clarify in the manuscript. Moreover, we acknowledge that we do not correctly cite the related works to justify our approximations, we will add such justifications in the manuscript.
>
>  - **Gaussian approximation:** A substantial body of research has built upon this approximation and verified its validity ([1], [5], [10], [11], [12]). In particular [5] demonstrated how modelling the SGD noise by a Gaussian *is sufficient* to understand its generalization properties which is the main interest of our work.
>  - **First derivative constant:** This approximation allows us to disentangle the first derivative of the loss and the product of the gradients isolating the scale of the noise, this corresponds to the decoupling approximation introduced and empirically verified in [2] (Figure 1, page 10). To further verify this assumption in our setting, we perform additional experiments comparing the spectrum of the SGD covariance with our approximations at stabilization. The results illustrate a substantial overlap in the spectrum (see attached pdf and point (3) in the main response) confirming the validity of our approximation.
>
> **Training Llama-3:** We agree that our study primarily focuses on small models like ResNets and GPT-2. Nevertheless, the primary aim of our work is to investigate the impact of weight decay when training models *from scratch*, for this reason, and given our computational resources, it would be infeasible to extend our experiments to significantly larger models such as Llama-3.
>
> **Other regularization methods:** Despite comparing with other regularization methods could give better guidelines for practitioners, the focus of this manuscript is not on understanding which regularization method works better but rather on providing a specific understanding of the mechanisms through which weight decay improves the performance of models trained in two main settings. Therefore, we did not compare with other regularization methods such as dropout because it would not provide further insights regarding the functioning of weight decay. Regarding batch normalization, we would like to point out that most of the experiments are conducted with architectures that have a normalization layer (BatchNorm for ResNets and LayerNorm for GPT2). Nevertheless, removing the normalization layers would be impractical, leading to exponential growth of the output variance (see [14]), therefore requiring additional modifications which would alter the consistency of the comparison. Moreover, in Appendix C.4, we report some experiments with *smaller* VGG networks without batch normalization and observe a similar dynamics when using weight decay.
>
> **Fixed hyperparameters:** For all main experiments, we have performed a thorough grid search over the learning rate and weight decay parameters (which are the main hyperparameters tuned in practice). E.g., see Figure 3 for the over-training regime and Figure 22 for the under-training regime. However, some other secondary hyperparameters are fixed because they do not alter the conclusions.
> For example, in the over-training setting, we fixed the coefficient $\beta$ of the EMA $EMA_t = \beta \cdot EMA_{t-1} + (1-\beta) \cdot w_t$ to be $\beta = 0.999$. For $\beta$ closer to 1, the effect is to make the EMA change slowly and smoother. This means that the EMA will incorporate a longer history of the parameter values and will be less influenced by short-term fluctuations therefore better approximating the mean of the stationary distribution. In our experiments we didn't observe any significant difference for $\beta \in [0.9,1)$
>
>
> Similarly for the fine-tuning, we observed that as long as the learning rate used in this phase is smaller than $10^{-3}$, we converge to solutions which have the same test error but for smaller learning rates longer time is needed. We are happy to provide further details about the effect of other hyperparameters.
>
> We thank the reviewer for pointing out the font size issue in the figures and the inconsistency in using "weight decay" versus "WD" in the main text. We will address these issues and make the necessary corrections in the manuscript.
>
> # Questions:
> 1. We answered the first part of the question above in the paragraph "**First derivative constant**". Given that we focus on standard classification tasks, could the reviewer elaborate on the meaning of multimodal data in this context? Generally, the i.i.d. assumption for the dataset is necessary for our approximations to hold.
>
> 2. In our work, we are considering multi-class classification over image datasets, such as CIFAR-10 and Tiny ImageNet. These datasets are inherently heterogeneous, comprising multiple classes. We hope this addresses your concern otherwise we would be happy to provide further details.
>
> 3. We suspect that the numerical instability comes from `bfloat16`'s limited precision, which can occur when different components in the network with varying scales are added together (e.g., values in the residual stream). Since the `bfloat16` data type has the same exponent size as `float32`, high-weight norm alone should not pose a problem. However, the precision of `bfloat16` is very limited, e.g., `bfloat16(256.0) + bfloat16(1.0)` is equal to `256.0` instead of `257.0`. We suspect this precision limitation is the primary problem in `bfloat16` runs without weight decay.

---

### Official Review · Reviewer_6pLz · 2024-07-15

**Soundness:** 3
**Presentation:** 3
**Contribution:** 3
**Rating:** 7
**Confidence:** 4

**Summary:**

This paper studies the effect of weight decay (WD) in the over-training and under-training regimes. It hypothesizes and empirically verifies different roles that WD plays in these scenarios. Specifically, in the over-training regime, WD primarily induces the implicit regularization of SGD via controlling the noise scale $\sigma_{\eta,\gamma}$. In the under-training regime, WD serves to stabilize and improve the optimization of the training loss as well as enabling better stability in low-precision training settings.

**Strengths:**

1. The paper studies WD, an important hyperparameter in the modern machine learning practice. Although widely used, understanding of its effectiveness is limited. This paper provides a rather comprehensive study on this problem.

2. By treating the over-training and under-training regimes separately, the paper uncovers the different roles of WD in model training. The results are insightful and could lead to future work.

3. The presentation is structured and mostly clear.

**Weaknesses:**

1. Some figures could be improved and(or) better explained. For example, Figure 2c presents results on the evolution of norm of weights and the correlation between the norm and the test error. It is quite difficult to cross reference which lines correspond to which test errors. It would be better if the authors could pinpoint specific lines in the figure or in the caption which lines one should look at. This would improve readability of the experimental results. Similarly for figures such as Figure 4.
2. EMA and finetuning would introduce additional hyperparameters (i.e. the momentum parameter and the smaller lr respectively). In the paper and appendix, I did not find their specific settings. The effect of these hyperparameters on the optimization is not, but should be, examined in the paper.

**Questions:**

1. The statement that EMA or fine-tuning “allows for the effective exploitation of the accumulated hidden regularization” is not entirely clear to me. Does generalization require a smaller scale of the stochastic noise and why? I feel this point is not adequately addressed or relevant prior work is not cited. In addition, if this is true, the current message seems to suggest that a large LR, combined with WD, is required to induce the stochastic noise but its scale has to be tuned down for better generalization. Is it not possible to induce the noise of appropriate scale at the very beginning with a careful selection of LR and WD?
2. Related to the previous question, some works actually use a warm-up schedule for the WD, together with LR decay(e.g. DINO: https://arxiv.org/abs/2104.14294). This would lead to a constant noise scale $\sigma_{\eta,\gamma}$. This seems to contradict with the previous statement about the necessity of a smaller noise scale. I am interested in learning about the authors’ thoughts about this.
3. The current work studies the training of networks from scratch. Would these conclusions/observations carry over to the fine-tuning of networks (e.g. LoRA)?

**Limitations:**

The authors note the limitations of the current work as the experiments reported are of smaller scale due to computational constraints.

---

> ### Author Rebuttal · Authors · 2024-08-07
>
> We thank the reviewer for their helpful comments, which will allow us to improve the manuscript.
>
> **Improved figures:** We thank the reviewer for suggesting improvements to our figures. We will make sure to reference specific lines in the text to improve the readability of the results.
>
> **Additional hyperparameters:** Some of the secondary hyperparameters are fixed because they do not alter the conclusions of our experiments. We fixed the cofficient $\beta$ of the exponential moving average $EMA_t = \beta \cdot EMA_{t-1} + (1-\beta) \cdot w_t$ to be $\beta = 0.999$. For $\beta$ closer to 1, the effect is to make the EMA change slowly and smoother. This means that the EMA will incorporate a longer history of the parameter values and will be less influenced by short-term fluctuations therefore better approximating the mean of the stationary distribution. In our experiments we do not see any significant difference for $\beta \in [0.9,1)$
>
> Similarly for the fine-tuning, in our experiments, we observed that as long as the learning rate used in the fine-tuning is smaller than $10^{-3}$, we converge to solutions which have the same test error but for smaller learning rates longer time is needed. We will better discuss these observations and provide further details regarding the choice of these hyperparameters in the manuscript.
>
>
>
> **Questions**:
>
> 1. The meaning behind the statement "EMA or fine-tuning allows for the effective exploitation of the accumulated hidden regularization" is discussed in more detail in Section 2.3. Specifically, we do not imply that generalization requires a smaller scale of noise throughout the entire training process. As shown in Figure 4, there is an optimal value for the scale of the noise. Our point is that to fully leverage the generalization benefits accrued from the noisy process, it is necessary to subsequently reduce the variance. To further clarify this point we can consider the red and green lines in Figure 2a. Without reducing the noise after a certain number of iterations, the test error remains high, around 35%. However, when using EMA or fine-tuning, we can observe the benefits of having a large noise level during the initial phase. This is because the implicit regularization of the Jacobian regulated by the noise scale affects the mean of the distribution and not the current interates. To realize these benefits, the noise level needs to be reduced to converge to the mean after this initial transient phase. Therefore, it is not possible to have an appropriate noise scale from the beginning that remains constant. The generalization benefits arise from starting with a large noise scale, which then needs to be appropriately reduced to converge to the mean.
>
>
> 1. After reviewing the implementation details of the referenced work, it appears that they indeed combine LR warm-up and decay with a cosine schedule but also apply decay to weight decay using a cosine schedule. Given that both learning rate (LR) and weight decay are decreasing along the training trajectory, this implies, according to our findings, that the scale of the noise should also decrease over time. Moreover, this aligns with our intuition that after the initial phase with a large noise scale, one should apply some variance reduction to fully exploit the generalization benefits.
>
>
> 1. Fine-tuning typically uses only a few epochs and LoRA restricts the model's capacity, so we believe its training regime is closer to undertraining. Thus, we believe that our main findings should transfer to the fine-tuning regime. However, we are not aware of any work that would notice a crucial effect of weight decay specifically for fine-tuning (assuming that the learning rate is carefully tuned), which is why we have not covered this topic.

---

> > ### Comment · Reviewer_6pLz · 2024-08-13
> >
> > Thank you for your responses and clarification. Overall, I find the paper and the additional experiments to be insightful. I will maintain my score and recommend for its acceptance.

---

### Official Review · Reviewer_YyLF · 2024-07-18

**Soundness:** 3
**Presentation:** 2
**Contribution:** 3
**Rating:** 6
**Confidence:** 2

**Summary:**

The paper explores the role of weight decay (WD) in training modern deep neural networks, focusing on its impact on optimization dynamics in two distinct training regimes: over-training (multiple passes through the dataset) and under-training (limited passes due to large dataset sizes and computational constraints).

In the over-training regime, the authors demonstrate that WD does not act as a capacity constraint but influences optimization dynamics by enhancing implicit regularization of SGD, as well as combines with large learning rates to maintain non-vanishing SGD noise, improving generalization by controlling the norm of the Jacobian.

For large language models (LLMs) trained with limited data passes, the authors discuss the relationship between WD and the effective learning rate, and how WD can prevent divergence in training with reduced precision, such as bfloat16.

**Strengths:**

1. This paper studies the role of weight decay in deep learning, moving beyond traditional views of regularization to consider its impact on optimization dynamics. By introducing the new perspectives of weight decay, the paper aids practitioners in better configuring their models(like the combination of learning rate and weight decay rate), potentially improving both training stability and model performance.

2. Extensive experiments on ResNet architectures and large language models on both multi-pass training and one-pass training validate the theoretical claims, demonstrating WD's practical benefits in both over-training and under-training contexts. The experimental results well validate the hypotheses introduced by the authors.

**Weaknesses:**

1. While the practical implications are well-explored, the theoretical explanations could be helpful. Some claims about implicit regularization and optimization dynamics are not fully explained.

2. The study focuses on specific architectures like ResNet-18. It would be beneficial to see similar analyses on a broader range of models to strengthen the generality of the conclusions.

**Questions:**

Please refer to weaknesses.

---

> ### Author Rebuttal · Authors · 2024-08-07
>
> We appreciate the reviewer's suggestion to enhance the theoretical explanations in our paper. We understand the importance of providing thorough theoretical insights.
>
> - **Conjecture:** We acknowledge that our results regarding implicit regularization are presented as conjectures. This decision was made consciously, as we believe it is challenging to rigorously prove these results within the scope of our current work. However, we included this conjecture to summarize our intuition and facilitate the understanding of our main message providing a conceptual framework that could guide future research.
>
>
>
> - **Optimization dynamics:** We have largely built upon the existing literature on label noise, as detailed in Section 2.2 of our paper. Previous studies have observed implicit regularization of the Jacobian in the presence of label noise. Building on this, we hypothesize that weight decay with large learning rates induces a similar implicit regularization effect, without any additional label noise but through loss stabilization mechanisms. We believe this perspective offers a novel viewpoint on how weight decay influences the optimization dynamics and enhances SGD implicit regularization.
>
>
>
> - **Additional architectures:** Regarding a broader range of models, in the appendix, we present similar experiments showing loss stabilization and better generalization due to weight decay for VGG and ResNet-34 architectures, along with fully scale-invariant ResNet architectures. For language models, the transformer architecture completely dominates the research field, therefore we considered it to be sufficient.

---

### Author Rebuttal · Authors · 2024-08-07

We thank the reviewers for their suggestions and feedback on our manuscript, which will help improve the quality and clarity of our work. We will incorporate your comments into the revised version of the manuscript.

We acknowledge that we did not adequately justify our approximations by citing the related work that introduces these approximations. Specifically, we approximate the noise of SGD with a Gaussian and make certain approximations on the covariance.
These approximations are widely recognized and employed in a substantial body of research. We would like to clarify the following points and provide additional experiments, which can be found in the attached pdf, to support our approach.

1. **Assumptions:** our series of approximations does not need to hold along the entire trajectory of SGD but only in the stabilization phase. This detail is important and we will clarify it in the manuscript.

2. **Negligible total gradient:** The term $L(\theta) \nabla L^\top(\theta)$ being negligible in equation 3 comes from the observation that the gradient noise variance dominates the square of the gradient noise mean. This fact has been used in previous works [2], [3], [4] and in particular [3] and [13] empirically verify it.

3. **Constant First Derivative:** The assumption of the first derivative being approximately constant across all data points, sometimes referred to as "decoupling approximation" [2], has been empirically verified for classification in [2]. Furthermore, we performed additional experiments to verify the decoupling approximation in our setting by comparing the spectrum of the SGD covariance with our approximations. In particular, we use Stochastic Lanczos Quadrature [15] to estimate the empirical spectral density of the SGD covariance with and without the decoupling approximation, during the large-LR phase. The experiments are performed for ResNet-18 trained on the cifar10 dataset for different combinations of learning rate and weight decay which are used in the manuscript. The results (reported in the attached pdf) illustrate a substantial overlap in the two spectra which serves as a validation of the reliability of our approximation.


4. **Gaussian approximation:** We agree that the validity of the Gaussian approximation for the noise of SGD has been long questioned. Nevertheless, a substantial body of research has built upon this approximation and verified its validity ([1], [5], [10], [11], [12]). In particular [5] demonstrated how modelling the SGD noise by a Gaussian *is sufficient* to understand its generalization properties which is the main interest of our work.

We will ensure these justifications and the additional experiments are added to the manuscript, with particular attention to the following references, which we report here to use in the responses below.

[1] Li, Z., Lyu, K., and Arora, S. "Reconciling modern deep learning with traditional optimization analyses: The intrinsic learning rate."

[2] Mori, T., et al. "Power-law escape rate of SGD."

[3] Zhu, Z., et al. "The anisotropic noise in stochastic gradient descent: Its behavior of escaping from sharp minima and regularization effects."

[4] Jastrzębski, S., et al. "Three factors influencing minima in SGD."

[5] Li, Z., Malladi, S., and Arora, S. "On the validity of modeling SGD with stochastic differential equations (SDEs)."

[6] Liu, Y., Yu, S., and Lin, T. "Regularizing deep neural networks with stochastic estimators of Hessian trace."

[7] Hoffman, J., Roberts, D.A., and Yaida, S. "Robust learning with Jacobian regularization."

[8] Sokolić, J., et al. "Robust large margin deep neural networks."

[9] Andriushchenko, M., et al. "SGD with large step sizes learns sparse features."

[10] Smith, S., Elsen, E., and De, S. "On the generalization benefit of noise in stochastic gradient descent."

[11] Xie, Z., Sato, I., and Sugiyama, M. "A diffusion theory for deep learning dynamics: Stochastic gradient descent exponentially favors flat minima."

[12] Li, Z., Wang, T., and Arora, S. "What happens after SGD reaches zero loss?--A mathematical framework."

[13] Saxe, A.M., et al. "On the information bottleneck theory of deep learning."

[14] Zhang, H., Dauphin, Y.N., and Ma, T. "Fixup initialization: Residual learning without normalization."

[15] Yao, Zhewei, et al. "Pyhessian: Neural networks through the lens of the hessian."

[16] Roburin, Simon, et al. "Spherical perspective on learning with normalization layers."

---

> ### Author Response · Authors · 2024-08-13
> **Additional experiment suggested by reviewer tBnp**
>
> To further validate our conjecture, we performed the experiment suggested by reviewer tBnp. Specifically, within the same ResNet18 on CIFAR-10 setting as in our main experiments, we created snapshot ensembles by averaging in function space along the SGD trajectory every 10 epochs for the combinations of learning rate (LR) and weight decay (WD) considered in the paper.
>
> To assess whether the mean of the stationary distribution in function space aligns closely with the EMA, where the Jacobian norm is regularized, we compared the performance of snapshot ensembles with that of the EMA. Additionally, we computed the Total Variation Distance (TVD) between the softmax outputs of the ensemble and the EMA on the Test set $TVD = \frac{1}{2N} \sum_{i=1}^{N} \sum_{j=1}^{C} \left| p^{(i)}_{\text{ensemble}, j} - p^{(i)}\_{EMA,j} \right|$
> . The results in **Table 1** show a strong alignment in test accuracies, while those in **Table 2** indicate a low Total Variation across all combinations. Together, these findings offer further validation for our conjecture.
>
> We believe these additional experiments contribute to strengthening our contributions. While time constraints limited us to running these experiments on CIFAR-10, we plan to extend them to TinyImageNet and include results from both datasets in the revised version of the manuscript.
>
>
>
> | WD     |**LR=0.001**|**LR=0.001**|**LR=0.005**|**LR=0.005**|**LR=0.01**|**LR=0.01**|**LR=0.025**|**LR=0.025**|**LR=0.05**|**LR=0.05**|**LR=0.1**| **LR=0.1**  | **LR=0.15** |**LR=0.15**|
> |--------|-----------|--------|-----------|--------|----------|--------|-----------|--------|----------|--------|---------|--------|----------|--------|
> |        | Ensemble  |  EMA   | Ensemble  |  EMA   | Ensemble |  EMA   | Ensemble  |  EMA   | Ensemble |  EMA   | Ensemble|  EMA   | Ensemble |  EMA   |
> | 0.0000 | 0.32      | 0.33   | 0.27      | 0.26   | 0.24     | 0.25   | 0.17      | 0.17   | 0.17     | 0.17   | 0.13    | 0.13   | 0.13     | 0.13   |
> | 0.0005 | 0.32      | 0.32   | 0.29      | 0.29   | 0.24     | 0.24   | 0.18      | 0.17   | 0.13     | 0.16   | 0.13    | 0.13   | 0.11     | 0.13   |
> | 0.0010 | 0.32      | 0.33   | 0.25      | 0.27   | 0.21     | 0.21   | 0.13      | 0.19   | 0.10     | 0.13   | 0.10    | 0.11   | 0.11     | 0.11   |
> | 0.0015 | 0.32      | 0.34   | 0.23      | 0.22   | 0.22     | 0.25   | 0.11      | 0.14   | 0.10     | 0.12   | 0.10    | 0.10   | 0.09     | 0.09   |
> | 0.0025 | 0.30      | 0.30   | 0.22      | 0.22   | 0.19     | 0.20   | 0.09      | 0.10   | 0.10     | 0.11   | 0.10    | 0.10   | 0.10     | 0.11   |
> | 0.0050 | 0.33      | 0.34   | 0.21      | 0.20   | 0.12     | 0.16   | 0.10      | 0.10   | 0.10     | 0.10   | 0.09    | 0.09   | 0.10     | 0.09   |
> | 0.0075 | 0.35      | 0.37   | 0.15      | 0.16   | 0.10     | 0.11   | 0.11      | 0.11   | 0.09     | 0.08   | 0.10    | 0.09   | 0.12     | 0.10   |
> | 0.0100 | 0.31      | 0.34   | 0.13      | 0.15   | 0.11     | 0.11   | 0.10      | 0.10   | 0.11     | 0.10   | 0.11    | 0.09   | 0.13     | 0.13   |
>
> **Table 1:** Test Error for Snapshot ensemble and EMA for different values of LR and WD
>
> | WD     | **LR=0.001** | **LR=0.005** | **LR=0.01** | **LR=0.025** | **LR=0.05** | **LR=0.1** | **LR=0.15** |
> |--------|--------------|--------------|-------------|--------------|-------------|------------|-------------|
> | 0.0000 | 0.03         | 0.02         | 0.01        | 0.01         | 0.01        | 0.01       | 0.01        |
> | 0.0005 | 0.04         | 0.02         | 0.02        | 0.03         | 0.09        | 0.08       | 0.07        |
> | 0.0010 | 0.04         | 0.05         | 0.04        | 0.10         | 0.09        | 0.07       | 0.07        |
> | 0.0015 | 0.04         | 0.07         | 0.07        | 0.08         | 0.08        | 0.07       | 0.07        |
> | 0.0025 | 0.04         | 0.11         | 0.10        | 0.06         | 0.08        | 0.08       | 0.09        |
> | 0.0050 | 0.06         | 0.15         | 0.11        | 0.09         | 0.08        | 0.10       | 0.11        |
> | 0.0075 | 0.08         | 0.12         | 0.10        | 0.10         | 0.10        | 0.11       | 0.12        |
> | 0.0100 | 0.10         | 0.09         | 0.10        | 0.10         | 0.11        | 0.12       | 0.13        |
>
> **Table2:** Total variation Distance between softmax output of Ensemble and EMA.
>
> [17] Huang, Gao, et al. "Snapshot ensembles: Train 1, get m for free."

---

### Decision · Program_Chairs · 2024-09-25

**Decision:**

Accept (poster)

**Comment:**

This paper studies the role of weight decay in the modern learning regimes of over-training computer vision models and under-training language models. The authors show that the weight decay does not act as a simple explicit regularizer, but as a component that affects the training dynamics in different ways for over-training and under-training.

This paper provides new insights that contribute to the ongoing efforts to explain the deep learning practice. The authors base their claims by extensive experiments and their explanations use nonrigorous mathematical formulations of the SGD dynamics. All of these, sufficiently support the authors’ claims.


The reviews reflect that the contributions of this paper are of sufficient interest and significance. The provided reviews are thorough, and the authors succeeded to improve their paper during the rebuttal and discussion with the reviewers. Accordingly, Reviewers S7kS and tBnp increased their review scores. It is worth to elaborate here on the following points that appeared in the Authors-Reviewers discussion and later in the AC-Reviewers discussion:
1. *Actionable insights and guidelines:*  This paper adds to the foundational understanding of the deep learning practice. Therefore, the understanding gained from this paper can help experts in machine learning and optimization to further improve the practical guidelines related to weight decay. In addition, the authors' results add to the existing literature by further emphasizing the role of the product of the weight decay and learning rate hyperparameters, which is relevant for tuning these hyperparameters in practice. Following my discussion with the reviewers, I believe that these contributions are of sufficient interest for publication.
2. *Assumptions and approximations:*  While the reviews considered some of the mathematical approximations as potential weaknesses of this work, the authors emphasize in their rebuttal and discussion that the approximations need to hold only in the stabilization phase and not throughout the entire SGD optimization trajectory. This seems adequate, considering that this paper does not intend to provide a rigorous theory but to empirically explain the practice.


Based on reading the paper, the reviews, the authors’ rebuttal and their discussion with the reviewers, as well as my discussion with the reviewers, I find this paper of sufficient quality and significance for publication. My recommendation is therefore to accept this paper.